
# The Future of Earth Observation in Hydrology

Matthew F. McCabe[1], Matthew Rodell[2], Douglas E. Alsdorf[3], Diego G Miralles[4], Remko Uijlenhoet[5], Wolfgang Wagner[6,7], Arko Lucieer[8], Rasmus Houborg[1], Niko E.C. Verhoest[4], Trenton E. Franz[9], Jiancheng Shi[10], Huilin Gao[11] and Eric F. Wood[12]

[1] Water Desalination and Reuse Center, King Abdullah University of Science and Technology, Thuwal, Saudi Arabia
[2] Hydrological Science Laboratory, Goddard Space Flight Center (GSFC), National Aeronautics and Space Administration (NASA), Greenbelt, Maryland, United States
[3] Byrd Polar and Climate Research Center, The Ohio State University, Columbus, Ohio, USA
[4] Laboratory of Hydrology and Water Management, Ghent University, Ghent, Belgium
[5] Hydrology and Quantitative Water Management Group, Wageningen University, The Netherlands
[6] Department of Geodesy and Geoinformation, Technische Universität Wien, Austria
[7] Center for Water Resource Systems, Technische Universität Wien, Austria
[8] School of Land and Food, University of Tasmania, Hobart, TAS 7001, Australia
[9] School of Natural Resources, University of Nebraska-Lincoln, Lincoln, NE 68583, USA
[10] State Key Laboratory of Remote Sensing Science, Institute of Remote Sensing and Digital Earth, Chinese Academy of Sciences and Beijing Normal University, Beijing, China
[11] Zachry Department of Civil Engineering, Texas A&M University, College Station, TX 77843, United States
[12] Department of Civil and Environmental Engineering, Princeton University, Princeton, New Jersey, USA

*Correspondence to*: Matthew F. McCabe (matthew.mccabe@kaust.edu.sa)

**Abstract.** In just the past five years, the field of Earth observation has evolved from the relatively staid approaches of government space agencies into a plethora of sensing opportunities afforded by CubeSats, Unmanned Aerial Vehicles (UAVs), and smartphone technologies that have been embraced by both for-profit companies and individual researchers. Over the previous decades, space agency efforts have brought forth well-known and immensely useful satellites such as the Landsat series and the Gravity Research and Climate Experiment (GRACE) system, with costs typically on the order of one billion dollars per satellite and with concept-to-launch timelines on the order of two decades (for new missions). More recently, the proliferation of smartphones has helped to miniaturise sensors and energy requirements, facilitating advances in the use of CubeSats that can be launched by the dozens, while providing 3-5 m resolution sensing of the Earth on a daily basis. Start-up companies that did not exist five years ago now operate more satellites in orbit than any space agency and at costs that are a mere fraction of an agency mission. With these advances come new space-borne measurements, such as high-definition video for understanding real-time cloud formation, storm development, flood propagation, precipitation tracking, or for constructing digital surfaces using structure-from-motion techniques. Closer to the surface, measurements from small unmanned drones and tethered balloons have mapped snow depths, floods, and estimated evaporation at sub-meter resolution, pushing back on spatio-temporal constraints and delivering new process insights. At ground level, precipitation has been measured using signal attenuation between antennae mounted on cell phone towers, while the proliferation of mobile devices has enabled citizen-science to record photos of environmental conditions, estimate daily average temperatures from battery state, and enable the measurement of other hydrologically important variables such as channel depths using commercially available wireless devices. Global internet access is being pursued via high altitude balloons, solar planes, and hundreds of planned satellite launches, providing a means to exploit the Internet of Things as a new measurement domain. Such global access will enable real-time collection of data from billions of smartphones or from remote research platforms. This future will produce petabytes of data that can only be accessed via cloud storage and will require new analytical approaches to interpret. The extent to which today's hydrologic models can usefully ingest such massive data volumes is not clear. Nor is it clear whether this deluge of data will be usefully exploited, either because the measurements are superfluous, inconsistent, not accurate enough, or simply because we lack the capacity to process and analyse them. What is apparent is that the tools and techniques afforded by this array of novel and game-changing sensing





platforms presents our community with a unique opportunity to develop new insights that advance fundamental aspects of the hydrological sciences. To accomplish this will require more than just an application of the technology: in some cases, it will demand a radical rethink on how we utilise and exploit these new observation platforms to enhance our understanding of the Earth system.

**1    Introduction**

The capacity to observe the hydrosphere from the vantage point of space has redefined not only our perspective of our planet as an interconnected system, but also how we describe the dynamic processes that occur above, on and beneath its surface. 2017 marks the 60[th] anniversary of the launch of Sputnik 1, a polished metal sphere less than 60 cm in diameter that became the first man-made object placed into orbit. Although only broadcasting dual-frequency radio transmissions over a short 21-

day period (until the batteries ran out), Sputnik had an indelible impact on humanity's perception of space, triggering the "space-race" and heralding in a new era of Earth observation (EO). Space was to become the new frontier. While the earliest satellite systems had a military reconnaissance focus, the value of space-based sensors for monitoring weather and climate was quickly recognised (Singer, 1957). Several meteorology focused systems were launched in the years following Sputnik, including the Television and InfraRed Observation Satellite (TIROS 1) in 1960, Nimbus 1 in 1964 and the Environmental

Science Services Administration (ESSA-1) satellite in 1966. However, it would be 15 years after Sputnik before the first civilian focused digital multispectral sensors were launched on-board the inaugural Landsat 1 mission in 1972, a program that has since continued uninterrupted for more than four decades (Wulder et al., 2008), providing an unrivalled record of terrestrial change and dynamics. Since these early satellite missions there have been considerable and dramatic advances in remote observation platforms and the types of measurements available from them. Evolving from early pan-chromatic and

red-green-blue (RGB) or R-G-near-infrared (NIR) imagery (De Wulf et al., 1990), sensor technology has expanded to include multi- and hyper-spectral visible to near-infrared bands (VNIR) (Houborg et al., 2015), multi-band thermal (Roberts et al., 2012), multi-channel microwave emissions (Njoku and Li, 1999), as well as radar and lidar techniques (Mace et al., 2009), all of which have advanced and redefined our knowledge and understanding of the Earth system.

From a hydrological sciences perspective, remote sensing has driven process insights and provided new and independent datasets that span a range of water cycle components. Recent studies such as Lettenmaier et al. (2015) provide a retrospective assessment of these developments and the progress of satellite observations in hydrology, complimenting earlier reviews of Schmugge et al. (2002) and Tang et al. (2009). In addition, process focused contributions have examined remotely sensed evaporation (Kalma et al., 2008; Wang and Dickinson, 2012), soil moisture (Njoku and Entekhabi, 1996;

Wagner et al., 2007), precipitation (Kidd and Huffman, 2011), surface waters (Alsdorf et al., 2007) as well as terrestrial water storage changes using more recent gravity-based methods (Rodell and Famiglietti, 1999; Swenson et al., 2003). Leveraging the spatial coverage of satellite data, a number of research efforts have also taken advantage of extended temporal sequences of satellite observations to compile long-term global datasets (Miralles et al., 2011; Liu et al., 2011a;



Beck et al., 2017). Such satellite-derived products provide an independent means of examining hydrological system dynamics and response (McCabe et al., 2005; Brocca et al., 2014), and offer the opportunity for an assessment of trends and variability in water cycle components (Zhang et al., 2016b; Kidd, 2001; Liu et al., 2012; Miralles et al., 2014).

Considering the multitude of discipline specific papers detailing remote sensing applications in hydrology that have been published over the last few decades[1], it is apparent that Earth observations have played an undeniable role in advancing the state of hydrological science. However, while reviewing this role is instructive and important, our intent here lies principally in forecasting the emergent opportunities that more recent and near-future observational developments might have in advancing and redefining our understanding of the terrestrial system and its interlinked processes. To do this requires

expanding our review beyond just space-based sensors, especially since satellite remote sensing represents just one aspect of EO. Indeed, some of the earliest attempts at mapping and monitoring the Earth surface were conducted from hot-air balloons, progressing to fixed wing planes and ultimately high-altitude reconnaissance aircraft such as Lockheed's U-2 "spy-plane" (the geopolitical consequences of which precipitated a tactical shift to space-based sensing and the "open-skies" policy, leading ultimately to the space-race). Interestingly, while EO developments have largely been defined by finer and

finer spatio-temporal resolutions or an increasing number of resolvable bands, we are also witnessing something of a devolution in the choices available from our observing platforms. That is, some of the earliest approaches (balloons, fixed-wing aircraft, etc.) are being reimagined through technological advances in system design, power management, autonomous operation and the accuracy of navigational controls and communication infrastructure.

The overriding intent and purpose of this work is to explore some of these emergent technologies and observational approaches, highlighting new and innovative sensing platforms that are either still reaching maturity in terms of their applications potential, or are yet to be fully embraced or even recognised by the larger user-community. Another is to motivate some discussion on how we, as a community, might better utilise available information and analytical resources, while also exploring the rapidly changing landscape of traditional space agencies in the light of recent commercial ventures

in space-based observation. At the least, this review will present to the reader details on the ever-increasing number of observational tools and techniques that have the scope and potential to deliver new and powerful insights to our discipline.

## 2   Review of Space Based Earth Observing Systems

The United States, European, Chinese, Japanese, Canadian, Indian and other national space agencies operate a large number of space-based observing systems, providing a diverse range of measurement types and/or spatial and temporal coverage to

the science community. Including the International Space Station (ISS) and satellites operated jointly between U.S. and

---

[1] A search on SCOPUS using the terms "remote sensing hydrology" returns over 4,180 unique contributions (Jan 31st, 2017).





international agencies, NASA alone has eighteen major Earth science missions currently in orbit, while the European Space Agency (ESA) has eleven EO missions currently in operation and a range of future satellites in advanced stages of planning and launch readiness. Other Earth observing instruments from various international agencies are on-board small satellites and CubeSats, as well as the ISS. The petabytes of data gathered by these missions have supported tens of thousands of

scientific investigations, practical applications, and breakthroughs in our understanding of the planet. There have been launch and instrument failures along the way, but the vast majority of large space agency missions have met their baseline objectives. On the other hand, while there are many examples of successful joint international missions, which have reduced the costs and risks for the contributing countries while increasing collaboration and data uptake, attempts to coordinate multi-platform observing systems in recognition of shared goals such as holistic water cycle measurement have been

ineffective. Obstacles include scientific competitiveness, technological secrecy and political considerations, differing visions and needs, and the lack of an authoritative coordinating organization. There remain considerable challenges and barriers to overcome before an holistic and collaborative EO strategy can be realised.

While EO in all its forms is the focus of this synthesis paper, it is worth overviewing the somewhat narrower perspective of

satellite based remote sensing, given its key role in advancing hydrological observations. As detailed in Fig. 1, the last few decades have seen the launch of thousands of satellites, with over 4,000 placed into orbit during this time. Of these, approximately 1,400 are operational, with the largest proportion comprising systems that form the backbone of the global communication network. Earth observing satellites, which include those operated by government, military, civilian and commercial sectors, comprise around 25% of the operational satellites in space, representing some 360 unique platforms.

While national governments operate the majority of the EO based systems (57%), recent years have seen an increasing number of both commercial and civilian platforms being launched: a trend that is expected to continue (see Sect. 4).

**Figure 1. The state of play in space today. Estimates are based on the Union of Concerned Scientists satellite database, updated from 30/6/2016 (see http://www.ucsusa.org/nuclear-weapons/space-weapons/satellite-database). In terms of the sectors operating**
**Earth Observing systems (right panel), another 5% include shared systems between those listed.**

## 2.1    Problems, Challenges and Knowledge Gaps

The past 25 years have seen astonishing advances in our ability to observe hydrological phenomena, driven in part by the maturation of satellite remote sensing, surging computing power and data storage capacity (Lettenmaier et al., 2015). Global

measurements of rainfall, soil moisture, snow cover, groundwater storage change, surface water elevation, and other water cycle variables could scarcely have been imagined when the race to space began in 1957, and the innovators and agencies who shepherded these developments deserve to be commended. Nevertheless, as detailed in Table 1 and discussed throughout this section, there remain critical gaps in our hydrological measurement and analysis capabilities. Snowfall, snow water equivalent, evaporation, deep soil moisture, groundwater depth and total storage, water consumption, and water quality





remain elusive targets, despite hopes that satellite missions to be recommended by the 2017 edition of the U.S. Decadal Survey for Earth Science and Applications from Space will address some of these retrieval challenges. As will be proposed herein, continuous, holistic water budget observation would be superior to the current paradigm of asynchronous measurement of individual variables. However, apart from requiring a paradigm shift in how we undertake much of our

research, achieving this also requires a breakthrough in observation cost efficiency, such as cheap, reusable rockets, or some other game-changing innovation. In Sect. 3, we explore some of these shortcomings and suggest improvements, highlight existing opportunities and identify some new innovations that may be on the EO horizon.

**Table 1. Hydrological variables and the current and planned satellite remote sensing missions that can be used to estimate them.**
**We note that this list is not necessarily comprehensive and that there are possible trade-offs between resolution and accuracy that are not explicitly accounted for in this list.**

To date, only a handful of terrestrial hydrology dedicated EO missions have been designed and launched by national and international space agencies. These were enabled by a shift towards more user-oriented missions over the last two decades,

which allowed scientists to press for their data needs and help steer missions from their earliest design (Lettenmaier et al., 2015). As a result of this engagement, there has been an increase in the range of hydrological variables that can be retrieved from space, spanning far beyond the snow cover extent, land cover and topographic products of early satellite remote sensing research. Key elements of catchment and continental water balances are now routinely derived from the available suite of EO satellites. While this unprecedented wealth of data has brought about major advances in the study of large-scale hydrology,

there remain gaps that need to be filled to further advance our understanding of the hydrosphere, as well as challenges that need to be overcome to ensure progression in our system knowledge. Here we expand upon some of these, with the aim of providing context for many of the new techniques and observation platforms on the EO horizon (see Sect. 3).

1.  A fundamental challenge in EO are the limitations imposed by only measuring the spectral signature of solar, Earth
emitted and reflected radiation, and using this information to retrieve desired geophysical parameters (with GRACE being an exception). This challenge can be extended to a perception in the literature that geophysical variables are directly obtained from EO, whereas the reality is that complex retrieval models, with their various assumptions, simplifications and parameters, and non-unique solutions, are almost always used to transform the radiative measurements into geophysical variables. For some variables, this conversion is quite straightforward (e.g. NDVI), while
for others, the retrieval model may have underlying assumptions or require ancillary data that contribute significantly to the overall error in the final product (e.g. soil moisture, evaporation). Bearing in mind that the utility of Earth observations lies not just in their capacity to reveal insights on the hydrological cycle but also in their potential to benchmark the climate models that we use to project hydrological response, an important issue emerges when one considers the dependency of Earth observation datasets on models themselves. In some cases, the retrieval and climate



models that need to be benchmarked may even share model assumptions or ancillary data, and in other cases this interdependency can extend to the use of model climatologies (Liu et al., 2011b) or even reanalysis forcing data (Mueller et al., 2013) in the generation of EO-based datasets. Given this, it becomes critically important that a full description and accounting of both the model and the component forcing that underlie the production of EO data is provided.

2. A second challenge relates to a range of engineering and operational issues, including: a) if a satellite is to rotate around Earth at the same speed that the Earth rotates around its axis, then it must be placed above the equator in a geostationary position, approximately 35,786 km above mean sea level. At that altitude, and with current technologies, visible and NIR radiative frequencies can be measured at high temporal resolutions (minutes), but only at spatial scales on the order of

kilometres (GOES-R can be tasked to capture sub-areas of a full-disk at a frequency of 30 seconds). Lower altitude (~700 km) satellites generally operate in polar orbits, allowing them to image the entire Earth surface, but at a coarser temporal resolution of one to several days. Such orbital limitations impede the ability to observe fast weather and hydrologic processes over the diurnal cycle at the needed high spatial resolutions (sub- to 10's of meters). One way to leverage the higher-resolutions achievable from lower-orbits and overcome the temporal repetition (cadence) issue, is through the use

of more than one satellite i.e. constellations: a topic that is explored further in Sect. 3.4; b) related in part to the satellite orbital characteristics, sensors and platforms are often poorly-designed to provide data over regions where hydrological observations are the scarcest and most needed (e.g. tropics, poles, mountainous regions and urban areas), which limits the potential to close the hydrological balance at continental scales and advance our understanding of these processes; c) finally, many hydrological variables require observations in the microwave portion of the spectrum (from Earth emitted

radiation), but current technology limits antenna size and therefore spatial resolution (excepting synthetic aperture radar) and impedes the mounting of microwave sensors in geostationary satellites. Increased spatial resolution is necessary to help disentangle Earth emissions from heterogeneous land and atmospheric conditions (cloud and moisture variability, wet and dry surface areas, different vegetation classes, effects of topography, etc.).

3. A number of unsuccessful missions (e.g. OCO, Glory, Landsat 6) have demonstrated the often capricious nature of space based observation. Such mission failures (or instrument failures, as in the case of the SMAP radar) highlight that even with massive investment and allocation of resources to satellite programs, there is no guarantee of mission success, reflecting an inherent risk of single instrument platforms. Indeed, this third challenge relates to the scientific community's penchant to focus on using a single sensor to retrieve a single geophysical variable. Such 'stove-piping' of

both science teams and retrieval algorithm development has impeded the progress of more comprehensive approaches to estimating global-scale hydrologic datasets. Indeed, there are numerous satellite systems that are currently in orbit, or in advanced stages of planning, that we seem ill-prepared to exploit (e.g. hyperspectral sensing), while at the same time, we rely on other sensors or variables that are often used well beyond the intent or purpose for which they were designed (e.g. NDVI). This may be a consequence of too much data, and too little cross-disciplinary interaction. Either way, the result is

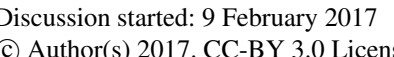



a plethora of variables that are being routinely collected by satellite systems, but which remain largely under-utilised by the community. As a manifest illustration of this issue, the current international Programs of Record (POR) includes over 700 existing or planned (approved) sensors for EO. It is more likely than not that most investigator or operational programs will use only a few of these.

4.  In terms of using data to advance our scientific understanding, multi-decadal observational records provide an ability to: (a) study trends in the terrestrial water cycle, (b) improve model simulations via e.g. data assimilation, and (c) benchmark the hydrology in the land-surface and atmospheric models that are being used to project the impact of climate and land use change on the water cycle. The development of long-term records demands the continuity (not necessarily

replication) of previous successful missions, which for space agencies with static budgets, may come at the expense of more innovative exploratory missions. Given the limited lifetime of satellite missions, these long-term records can only be achieved by merging datasets based on various sensors. As a consequence of this merging, observed inter-annual fluctuations may reflect discontinuities in the constellation of satellites, rather than actual hydro-climatological signatures. Further efforts to harmonize satellite data are required for the effective data assimilation or direct use of Earth

observations in hydrological models. Likewise, to support such efforts, information on the accuracy of hydrological retrievals is required. To do this will require a departure from simple sensor precision and ground validation statistics, towards more appropriate error analysis and statistical equivalence that can reflect the artefacts of multi-sensor merging strategies.

5.  Finally, although the number of EO systems available for hydrological monitoring seem to be increasing, one of the more concerning aspects threatening the very foundation upon which much of our process understanding and conceptual developments derive, is the decline of *in situ* networks, especially since the 1980's (Fekete et al., 2012). Distinct from the issue of poor spatial representation of ground-based monitoring that discriminates collections in the developed versus developing nations, this is a negative trend that has been replicated across many regions of the world, from the United

States (Lanfear and Hirsch, 1999) to the pan-Arctic (Shiklomanov et al., 2002). From a long-term monitoring perspective, one particularly worrying aspect of this decline is the demise of gauging stations (and other measurements) containing greater than 30 years of continuous records that has been witnessed in the US (see water.usgs.gov/nsip/history.html), but almost certainly seen in other parts of the world and for other hydrological variables (Lorenz and Kunstmann, 2012). Without long-term and well maintained *in situ* networks, the challenges of

disentangling the fingerprints of climate changes and its impact on hydrological and related systems becomes even more difficult (Hidalgo et al., 2009). While there have been encouraging activities that draw focus to the importance of *in situ* collections at catchment, regional and even global scales (Zacharias et al., 2011; Dorigo et al., 2011; Stahl et al., 2010), sustained community effort is required. The importance of a robust and operational *in situ* network is an often under-recognized element of satellite research programs and initiatives. Indeed, it is not outrageous to posit that there are few



conceptual advances or process insights resulting from space-based observations that have occurred independent of using ground based monitoring.

To resolve many of the above-mentioned challenges, we need to develop comprehensive programs that conceive of EO as being based upon a variety of complementary platforms (i.e. satellite arrays that include nano-satellites, commercial aircraft based sensors, long-deployment UAVs, high-altitude balloons, etc.; see Sect. 3) blended and merged with models in ways that are more informative than current data assimilation approaches. The community has already developed hyper-resolution land surface models that have been applied at 30 m scales over continental domains (Chaney et al., 2016), as well as approaches for merging land surface models with satellite retrievals to obtain time consistent data sets (Coccia et al., 2015), or using diverse data to challenge hydrological simulations (Koch et al., 2015; Stisen et al., 2011). Other approaches beckon, especially the opportunities being facilitated by cloud computing and data analytic techniques (see Sect. 3.8). The emerging hyper-resolution trend (Wood et al., 2011; Bierkens et al., 2015) will require hyper-resolution forcing data together with observations of the diurnal cycle of critical hydrological variables in order to prevent spatial and temporal inconsistencies between observations and models: a demand that we seem ill-prepared to meet. Sustained advances are needed to overcome these challenges. To do so, we have to recognize and accommodate the physics of EO, space agencies need to invest in new technologies (e.g. the development of nano-satellites and next-generation antenna), and they will also need to support the community to develop improved retrieval models and encourage the use of measurements from a variety of sensors. All of this will require open and easily accessible data systems, something that to date has not been streamlined or optimised in the most efficient manner. What emerges from this brief summary is that it is not necessarily any technological limitations that are inhibiting progress or advances. The challenges as listed seem largely scientific in nature, and reflect the need for a paradigm shift in how EO data is collected, disseminated and utilised, which is a topic that is examined further in Sect. 3.

### 2.2 Hydrology Specific Data Needs

Some of the issues identified in the above paragraphs are general to Earth observation as a discipline, rather than specific to the field of hydrology. For this reason, we shift the discussion to focus on some of the key data needs and knowledge gaps per water cycle variable. While the following is focused on satellite-based retrievals and does not explicitly detail other observation systems such as balloons, aircraft etc. the issues are not platform specific. In compiling this (deliberately concise) list of variable related concerns, we perpetuate a previously recognized limitation of our communities approach to EO i.e., the fixation on single component retrieval, whereby we measure one water cycle variable at a time, and ignore the interdependencies and relationships inherent in observed responses (López et al., 2016). Here, by at least acknowledging the problem, we seek to excuse ourselves from perpetuating it.

**Precipitation**. Satellite retrieved rainfall was first inferred using visible and thermal infrared observations (Lethbridge, 1967), providing an estimate of rainfall volume (Kidd and Huffman, 2011). With the launch of microwave sensors such as





the Advanced Microwave Sounding Unit-B (AMSU-B) and the Special Sensor Microwave Imager (SSMI), a shift towards more direct measurements of precipitation rates was taken. The evolutions in these early missions are reflected in the dedicated Tropical Rainfall Measuring Mission (TRMM) and the latest Global Precipitation Mission (GPM), both of which use a combination of radiometers and high-resolution radar measurements. However, precipitation is highly variable in both

time and space, so accurate representation demands a platform that matches these spatio-temporal constraints. Unfortunately, while advances have been made, current capability falls short in this regard. Microwave instruments on low Earth orbits limit repeat overpasses to once per day or longer. Although algorithms have incorporated infrared retrievals from geostationary satellites to fill the temporal gaps, these can introduce considerable uncertainty. Furthermore, measurement resolutions over land are typically greater than 5 km x 5 km, and are not anticipated to improve dramatically after GPM, presenting a further

challenge for those seeking for hyper-resolution hydrological modelling. So, while remote sensing observations of precipitation have greatly improved our understanding of its global magnitude and variability, there remain critical knowledge gaps in accurately characterizing this flux. Other long-standing issues include the detection of snowfall, drizzle and extreme events (Rios Gaona et al., 2016) and the fact that lower-frequency microwave channels often fail to discriminate between the scattering by ice in the clouds and that by the surface, while higher frequency channels require the extraction of

the background emission, which is not trivial. An additional challenge is rainfall retrieval at higher latitudes, where snowfall is the largest contributor to annual precipitation, but which remain unsampled by the oblique orbits of past TRMM or current GPM missions. All of these issues are amplified by the disconnect between satellite data and more traditional gauge-based measurements, which have well recognized problems of poor distribution, wind-induced under-catch, elevation bias in gauge placement, along with numerous other measurement complications (Lorenz and Kunstmann, 2012; Steiner et al., 1999).

While ground-based Doppler radars offer high temporal and spatial resolutions, they are mostly only available in developed countries (Heistermann et al., 2013), have relatively poor coverage in mountainous areas, and their merging with satellite observations has often proven cumbersome (Lee et al., 2015). Nevertheless, due to their spatial and temporal continuity, Doppler radar data are often considered the gold standard where they do exist and continued expansion of Doppler networks in the developing world is certain to benefit hydrology. However, efforts to archive, harmonize, reprocess and provide access

to the global network of radar data are needed before the potential value of this data source can be realized.

**Evaporation.** Monitoring the second largest flux in the continental hydrological cycle has proven to be especially challenging, whether from ground-based approaches or from space. Currently, a range of techniques to derive evaporation from remote sensing data exist (Anderson et al., 2004; Wang and Dickinson, 2012; Ershadi et al., 2014; Fisher et al., 2008).

However, given the inability to observe this water flux in any direct way, all such approaches rely on rather complex empirical or process-based models, often requiring significant ancillary information and site specific parameterisations. Moving forward, improvements are needed in both retrieval algorithms as well as satellite measurements. Large scale satellite-based evaporation estimates generally have a resolution that is too coarse for critical applications such as drought assessment, water management or agricultural monitoring, although there are regional to local scale exceptions to this

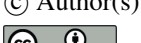



(Anderson et al., 2013; Cammalleri et al., 2014). To achieve the required high resolutions over large spatial domains, equally high-resolution observations of surface-level temperature and radiation budget components are required, together with improved representation of the hydrometeorology required to force many of these models. While current generation geostationary satellites can provide retrievals between 1-2 km in the visible-to-infrared spectrum, available global
operational data products only offer coarse degree-scale resolutions, presenting a critical drawback for efforts targeting the production of global evaporation estimates (McCabe et al., 2016; Miralles et al., 2016). One of the key issues to advance the development of evaporation models is our representation of the vegetation components inherent in partitioning between evaporation and transpiration (Ershadi et al., 2015). The emergence of new remote sensing data sets (see Vegetation section below), beyond the relatively simplistic NDVI or LAI approaches that are currently employed, may provide a path forward
to achieving these needed model improvements.

**Soil moisture.** Over the years, several algorithms have been formulated to derive soil moisture from low microwave frequencies, resulting in numerous data products being developed since the late 1970's. These datasets have demonstrated their usefulness for hydrological applications at different scales, and have become a valuable tool for the climate community
after their merging into multi-decadal, -satellite and -sensor (active and passive) records (Liu et al., 2012). With algorithm developments for the active scatterometer based retrievals (Naeimi et al., 2009) enabling soil moisture products from the Advanced Scatterometer (ASCAT) (Wagner et al., 2013), together with the launch of soil moisture dedicated missions such as the Soil Moisture and Oceans Salinity (SMOS) mission in 2009 and the Soil Moisture Active Passive (SMAP) mission in 2015, the retrieval of soil moisture from space has taken on a new dimension. Current research strives to improve the
accuracy of retrieval algorithms (Mladenova et al., 2014), understand the spatial representativeness of the observations (Dorigo et al., 2015), increase the spatio-temporal resolution (Jha et al., 2013; Merlin et al., 2010), optimally ingest observations into hydrological models (Reichle et al., 2007), and explore the blending of different sensors (Liu et al., 2011b). The coarse resolution of passive based retrievals remains a challenge, but advances in antenna technology may provide improvements on this. Likewise, shallow retrieval depths limit the determination of root-zone moisture profiles and
dynamics, although modelling approaches seek to improve deeper-soil representation (Das and Mohanty, 2006; Li et al., 2010). Despite the failure of the SMAP radar after only three months of operation, the performance of SMAP's passive retrievals has recently been evaluated with encouraging results (Pan et al., 2016). Likewise, the SMOS mission continues to provide impressive insights and an expanding range of derived products (Mecklenburg et al., 2016). Although no SMOS follow-on mission is planned at this time, the future of satellite remote sensing of soil moisture remains bright, with the
newly launched Sentinel-1 series from the European Space Agency (ESA) carrying high-resolution radars that have proven capabilities to deliver soil moisture at less than 1 km resolution and near-real time (Paloscia et al., 2013).

**Runoff.** Of all the hydrological variables, the one that typically draws the most attention from a water management perspective is river runoff. However, runoff is inherently local and difficult to determine from coarse-resolution space





observations. While some efforts have focused on using GRACE to derive long-term mean discharge for large rivers, most initiatives to date have been limited to running hydrological models of different complexities using satellite-derived digital elevation models, river height, inundation extent or simply satellite-based precipitation. The long-awaited Surface Water and Ocean Topography (SWOT) mission (now planned for 2021) is set to measure surface water bodies and to infer river

discharge. SWOT will carry a radar altimeter capable of deriving two-dimensional images of surface water height, with a vertical accuracy of about 1 cm averaged over 1 km$^2$ across a 120 km swath. This will deliver a substantial advance over previous altimeters used for hydrological applications that report only one-dimensional heights (Calmant and Seyler, 2006). However, while river height, width and slope will be derivable from SWOT, the calculation of river discharge will still rely on algorithms that account for the unknown channel depths and flow velocities. Any algorithm that has a requirement of *in*

*situ* data for calibration limits its applicability in ungauged regions, where discharge measurements from space are the most needed. Moreover, estimates of discharge will correspond to the particular time of the SWOT overpass, which may not match the desired timing, especially in applications related to the detection and monitoring of flash floods that require both high spatial and temporal resolution.

**Groundwater and terrestrial water storage.** Gravimetric remote sensing represents one alternative to conventional electromagnetic sensing techniques for estimating water storage variables. Since 2002, GRACE has been measuring temporal anomalies in the Earth's gravity field, from which changes in terrestrial water storage (the sum of groundwater, soil moisture, surface water, and snow) can be inferred (Tapley et al., 2004). Combined with auxiliary model- or observation-based information, satellite gravimetry provides the only viable remote sensing approach for consistently estimating changes

in groundwater storage (Rodell et al., 2007). However, GRACE's coarse spatial (>150,000 km$^2$) and temporal (monthly) resolution and data latency (typically 2-4 months) have limited its value for operational applications and decision-making, absent model-based downscaling (Zaitchik et al., 2008). The GRACE Follow-On mission (to be launched in 2018) is expected to improve upon the retrievable resolution somewhat (>100,000 km$^2$). Regardless of these limitations, the GRACE mission has proven to be one of the outstanding examples of non-traditional EO applications in hydrology, and serves as a

reminder that process understanding is best achieved utilising a range of complementary observation platforms.

**Vegetation.** Given the strong links between vegetation and multiple elements of the water cycle, there is understandable focus from the hydrological community on capturing plant response and dynamics at high spatial and temporal resolutions. Vegetation features are most clearly extracted from the VNIR, with sensors such as MODIS and Sentinel-2 providing

unprecedented detail on plant-spectrum response. It could be argued that too much emphasis has been placed on relatively simplistic broadband-derived optical or near-infrared vegetation indices such as the Normalized Difference Vegetation Index (NDVI), at the expense of other indices and portions of the electromagnetic spectrum (Houborg et al., 2015). For instance, microwave observations of Vegetation Optical Depth (VOD) offer a close proxy of the water content and hydrological functioning of vegetation (Liu et al., 2011a; Liu et al., 2013), without the impacts of signal saturation in dense canopies or





sun-sensor geometry issues. While research efforts have produced long-term records of VOD that hold considerable potential to improve understanding of land water fluxes and carbon storages (Liu et al., 2015), they are seldom employed in diagnostic studies of the hydrological cycle. In some ways, there seems to be a disconnect between the vegetation and water research communities that has led to the mis- or under-use of observable vegetation metrics. Solar induced fluorescence (SIF) is one

example of this disconnect (Meroni et al., 2009). Observations of fluorescence by the Japanese Greenhouse gases Observing SATellite (GOSAT) have mapped photosynthesis at the global scale (Frankenberg et al., 2011). Due to the synchronization of photosynthesis and transpiration through the stomatal conductance, SIF data could in principle be utilised to enhance our understanding of transpiration and evaporative stress (Alemohammad et al., 2016), but relatively little research has focused on examining this apparent link. Data from the Orbiting Carbon Observatory-2 (OCO-2) spectrometer, and the

TROPOspheric Monitoring Instrument (TROPOMI) on-board Sentinel-5 Precursor (to be launched in 2017), will enable some of these ideas to be explored further, forerunning the first SIF-dedicated mission, the FLuorescence EXplorer (FLEX) from the European Space Agency (ESA) (launch scheduled in 2022). While earlier SIF datasets had resolutions that were not particularly well suited for hydrological applications, OCO-2 and TROPOMI present improved spatial detail (3 km and 8 km, respectively), and in the case of TROPOMI, a frequent revisiting time (~ daily).

**Snow and permafrost.** Terrestrial snow and frozen soils represent an important yet poorly-represented component of the global water cycle. While the retrieval of two-dimensional snow cover extent is a mature research field (Hall et al., 1995), the retrieval of snow depth, density or water equivalent (SWE) is usually of greater interest to hydrologists, since these form key elements of model initialisation and forecasting of runoff, drought and flood prediction (Bormann et al., 2013).

Unfortunately, retrieving these and related cryospheric variables remains a major challenge, particularly for mountainous regions, where spatial variability is high and seasonal snow depth may reach tens of meters. Microwave sensors can be used for SWE and snow depth observations, but current systems lack optimal combinations of frequencies and resolutions. Although active microwave sensors can improve retrieval resolution and may be better suited for snow monitoring in mountainous regions, the maturity of active-based products has not reached the same level as passive approaches. However,

even passive microwave-based SWE retrieval can suffer from signal saturation due to a deep snowpack, with commonly used Ku- and Ka-band microwave emission signals saturating at around 200 mm SWE. Given the importance of monitoring wet-snow properties for hydrology, synthetic aperture radar (SAR) retrieval approaches have also been proposed, since passive approaches are not sensitive to dry snow parameters. Gravimetric techniques represent another alternative to microwave based measurement approaches for snow depth (Baur et al., 2009), but the approach is limited by the large spatial

and coarse temporal characteristics of such sensors (Niu et al., 2007). In light of the non-selection of ESA's CoreH2O as an Earth Explore mission, there remains a need for high-resolution active microwave sensors with high revisit times to more effectively capture the dynamics of wet-snow in diverse terrain. Apart from snow covered surfaces, understanding the dynamics of frozen soils has become an increasingly important topic in hydrology, given the observed warming in many cold-area regions and the role that permafrost may play in changing river discharges, particularly in Boreal regions (Woo et



al., 2008). Permafrost properties include key state variables such as ground temperature, as well as thickness of the active layer, spatial patchiness and ice content. While there has yet to be a dedicated permafrost mission, EO data can be used to obtain permafrost-related features, such as the evolution in micro-topography, rock glaciers, thermokarst and deformation. For instance, ESA's SMOS satellite has been used to detect the onset of soil freezing (Rautiainen et al., 2016) with

encouraging results, while the SMAP mission would have provided key insights into permafrost processes, particularly the freeze/thaw state, which acts as a proxy for monitoring methane and carbon release (Heimann and Reichstein, 2008), forest productivity (Kimball et al., 2001) and sub-surface flow processes (Bayard et al., 2005). ESA's Sentinel 1 mission (Sabel et al., 2012), together with InSAR data (Liu et al., 2010), may present as possible platforms from which permafrost characteristics can be retrieved, advancing our knowledge of this increasingly important variable and our understanding of

cold regions hydrology.

**Water vapour.** A general drawback of current satellite observations for hydrological applications is their inability to provide vertical profiles of the atmospheric state with a high enough temporal resolution to allow tracking of the fate and transport of water vapour. Water dynamics in the lower atmosphere are determined by complex interactions between the land surface and

the free atmosphere, which are mediated by the diurnal cycle of the atmospheric boundary layer (ABL, from the surface to 2–5 km above). Diurnal processes like air entrainment into the ABL act as key drivers of evaporation, convective rainfall or near-surface humidity. Therefore, understanding the connection between the surface and atmospheric branches of the hydrological cycle relies on adequately monitoring heat and moisture exchanges in the ABL over large spatial domains. However, this requirement demands the availability of temperature and humidity observations at fine time steps (e.g. hourly)

and over high vertical and horizontal resolutions. Presently, low orbiting sensors capable of providing vertical information, such as the lidar in the Cloud-Aerosol Lidar and Infrared Pathfinder Satellite Observation (CALIPSO), the radar in CloudSat, or the hyperspectral sounder in the Atmospheric Infrared Sounder (AIRS), can only provide data at daily (or longer) temporal resolutions. Existing geostationary satellites, on the other hand, have frequent temporal sampling, but wider spectral bands and coarser vertical resolutions. Until these capacities are resolved in a single platform (or in a constellation

of smaller satellite systems) the lack of any high spatio-temporal resolution data to monitor the evolution of the ABL will continue to constrain our ability to monitor diurnal cycles of atmospheric water fluxes from space.

**Water quality.** Compared with other hydrology related variables, little focus has been directed towards characterising inland surface water quality from space. Changes in the Earth's natural environment, whether from global warming, land use/land

cover change, or other anthropogenic causes, can significantly deteriorate freshwater quality (Whitehead et al., 2009). Given the limited availability of *in situ* water quality measurements, remotely sensed datasets offer a means to fill this knowledge gap, with temperature, suspended sediment, dissolved organic matter and chlorophyll of particular interest. Lake and stream temperatures, which directly impact freshwater habitats, are very sensitive to climate changes (van Vliet et al., 2013). Although land surface temperature products generated from sensors such as MODIS and Landsat are quite well developed,

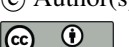



large-scale water temperature datasets are less common. While estimating water temperature requires delineating water bodies, a MODIS-based water mask has been developed for this purpose (Carroll et al., 2009). However, resolving such information remains challenging for water bodies with surface areas that have large seasonal or inter-annual variations, or whose cross-sections fall below retrievable resolutions. Suspended sediment and chlorophyll concentrations can be measured

using VNIR data from Landsat, Sentinel-3, MODIS, and AVHRR, or with hyperspectral sensors (Brando and Dekker, 2003). Research on the use of physically based algorithms that monitor these properties is required, with high spatial and spectral resolution observations needed to advance such efforts (Odermatt et al., 2012). The Hyperion mission has contributed to sensing such variables with high accuracy (Giardino et al., 2007), but will be decommissioned in late Feb, 2017. With no dedicated water quality mission planned, there is interest in the proposed Plankton, Aerosol, Cloud, and ocean Ecosystem

(PACE) satellite to advance terrestrial water quality monitoring, even with its primary focus on oceans.

Sect. 2 has focussed predominantly on our space-based observing systems and identified some of the issues hindering developments in our characterisation of the hydrological cycle. In order to ensure continued advances in our system understanding, it is paramount that we exploit a comprehensive and holistic EO strategy. To explore this concept and the

opportunities being provided by a combination of new technologies and analysis techniques, a presentation of emerging monitoring systems and approaches that may augment, support or even supplant the traditional notion of EO is presented in the following section.

### 3 Emergent Capability and Technologies

Fifty years from now, historians will hopefully regard our current remote sensing capabilities the way we regard

transportation in the early 20[th] century: most of the major modes were already in existence, but huge improvements in quality, cost and production efficiency, accessibility, and safety were yet to come. The improvements will be spawned and nurtured as before by government research investments, individual ingenuity, and private sector competition. In this section we briefly summarize near and mid-term plans of the space agencies and provide some focus on a range of recent innovations that will augment and possibly supplant the larger orbital missions in the near, middle, and long-term. Later, in

Sect. 4, we review the commercialization of space, which will be essential for driving down orbital insertion costs and thus enabling the predicted efficiency and accessibility improvements for many of the technologies described below. In previewing some of the observation platforms that will be presented, Figure 2 provides a concept of what a new Earth observing "System of Systems" would comprise.

**Figure 2. An Earth observing "System of Systems" for revolutionizing our understanding of the hydrological cycle. This multi-scale, multi-resolution observation strategy is not a concept: the technology exists and is largely in place now. Supporting traditional space based satellites, there are now a range of orbital options from commercial CubeSats to demonstration sensors on-board the International Space Station. Beyond orbiting EO systems, technological advances in hardware design and communications are opening the skies to stratospheric balloons and solar planes, as well as an explosion of UAV-type platforms**

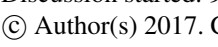


for enhanced sensing. At the ground level, the ubiquity of mobile devices are expanding traditional in-situ network capacity, while proximal sensing and signals of opportunity are opening up novel measurement strategies.

### 3.1    Future Agency Missions

In forecasting the range of future hydrology related satellite missions, it is not feasible to comprehensively list the entirety of national space agency plans in this brief overview. Realising this, we use US agency missions as guidance for comparable space programs in Europe, China and elsewhere. But, while the specifics might vary, there are some generalities that remain true. For example, space agencies typically discuss plans for their flagship Earth observing missions 10-15 years out, accept proposals and approve the formulation and science definition teams for missions 5-10 years out, and begin assembly 3-5

years out. While NASA Venture class and smaller missions as well as bolt-on instruments typically have more compressed timelines, it is clear that the time horizons from mission concept to launch are long, rather than short.

Some hydrology relevant flagship missions currently approved and in various stages of development include the GRACE Follow On, WCOM, SWOT and ICESat-2. The NASA/German GRACE Follow-On mission, with a launch window between

December 2017 and February 2018, will extend the unique monthly record of terrestrial water storage anomaly observations that have been provided by GRACE since 2002 (Tapley et al., 2004; Rodell et al., 2017). In addition to the K-band microwave ranging system used to measure changes in distance between its twin satellites with extreme precision, GRACE Follow-On will use an experimental laser ranging system and use design improvements, which together are expected to increase the spatial resolution from roughly 150,000 $km^2$ to 100,000 $km^2$ at mid-latitudes. China's Water Cycle Observation

Mission (WCOM), targeted for launch around 2020, aims to measure soil moisture, snow water equivalent, soil freeze-thaw, atmospheric water vapour, and precipitation, amongst other variables. This is to be accomplished through accurate, simultaneous active and passive microwave measurements across a wide frequency range, obtained by three on-board instruments: 1) an L-S-C tri-frequency Interferometric Microwave Imager with 15-50 km spatial resolution, consisting of a 9 x 6 m mesh reflector and a one-dimensional thinned array as the feed; 2) a Polarized Microwave Imager covering 7.2 to 90

GHz with a 1.8 m diameter reflector antenna for conical scan; and 3) an X-Ku dual-frequency polarized scatterometer with 2-5 km spatial resolution and 1000 km swath for snow water equivalent and freeze-thaw mapping. NASA's SWOT mission, scheduled for launch in 2021, will return accurate surface water elevations over 90% of the globe at least twice every three weeks, enabling estimation of river runoff as well as surface water storage. SWOT will employ a wide swath, Ka-band radar interferometer to resolve 100 m wide rivers and 250 $m^2$ lakes, wetlands, or reservoirs with a height accuracy of 10 cm and a

slope accuracy of 1 cm/km. Recent runoff data are currently available from only a fraction of the world's rivers due mainly to closed data policies outside of a few developed nations, thus SWOT will fill a major void in our observational capabilities. NASA's ICESat-2, while primarily focused on precise laser altimetry for ice sheet mapping, will also prove valuable for monitoring surface water elevations (Jasinski et al., 2016), particularly before the launch of SWOT. Other missions, such as


NOAA's Suomi National Polar-orbiting Partnership (Suomi NPP; launched in 2011) Joint Polar Satellite System 1 (JPSS-1; scheduled for launch near the end of 2017), and future missions in the JPSS series are mainly geared towards atmospheric measurements, but all (will) carry Visible Infrared Imaging Radiometer Suite (VIIRS) instruments, which collect visible and infrared imagery useful for monitoring snow cover and vegetation as an input to retrieval algorithms for numerous

hydrological variables. ESA's Sentinel-4 Earth observing mission (planned for launch in 2019) and its successor Sentinel-5 will focus on air quality monitoring. ESA also plans two Earth Explorer missions related to hydrology: 1) the Biomass mission (planned for launch in 2021) will carry a P-band synthetic aperture radar for the purpose of estimating forest biomass, but which may also be useful for inferring root zone soil moisture; and 2) the Fluorescence Explorer (FLEX) mission, which will map vegetation fluorescence to quantify photosynthetic activity and should help to constrain

transpiration rates. In addition to these large missions, NASA's Venture class ECOsystem Spaceborne Thermal Radiometer Experiment on Space Station (ECOSTRESS) (scheduled to be deployed aboard the ISS in 2018), will measure vegetation temperatures with the aim of constraining transpiration estimates and better understanding plant response to stress.

The 2017 edition of the Decadal Survey in Earth Sciences is intended to guide the prioritization and selection of major U.S.
Earth observing satellites for the next ten years. While the 2007 edition (National Research Council, 2007) recommended specific mission architectures, the new edition is expected to recommend observables and to leave mission and instrument design to the agencies and proposing institutions. It is unknown at the time of writing what hydrological observables will be prioritized, but based on the missions that were included in 2007 (but did not enter NASA's mission queue due to a second or third tier ranking), we speculate that snow water equivalent will be a priority. Referring to Table 1, evaporation is another

variable that may be targeted due to its importance, lack of a current, dedicated mission, and existence of a demonstrated retrieval approach. Deep soil moisture could also be on the list, although soil moisture algorithms that make use of wavelengths longer than L-band are less than mature.

While there are impressive and innovative sensing platforms scheduled for launch in the next 5-10 years (or in advanced
stages of planning) across international space and government agencies, there are emerging parallel opportunities for both investigator driven and commercially led activities that have the potential to reshape the EO landscape in hydrology, a selection of which are explored below.

### 3.2    UAVs, Stratospheric Balloons and Solar Planes

### 3.2.1    Unmanned Aerial Vehicles (UAVs)

One of the most exciting recent advances in near-Earth observation lies in the field of Unmanned Aerial Vehicles (UAVs), also referred to as Unmanned Aircraft Systems (UAS) or Remotely Piloted Aircraft Systems (RPAS). Often used interchangeably, or simply referred to as a drone, the terms encompass the remote or semi-autonomous operation of an





airborne vehicle. In a way, these new observation platforms represent a "hook in the sky" from which to deploy a range of sensors. The application of UAVs for remote sensing has offered new opportunities to map, monitor and understand the environment in unprecedented detail (Anderson and Gaston, 2013), particularly at the scale at which traditional field-based observations can be made, but also covering a greater spatial extent with a unique top-down view. The key advantages of

UAV-based remote sensing is their capacity to: 1) collect ultra-high resolution imagery (defined here as 1-20 cm pixel size) for mapping fine-scale features; 2) acquire data on-demand at critical times and with high temporal resolution at costs affordable to an individual investigator; 3) carry multiple sensors (both active and passive), across the electromagnetic spectrum; 4) be employed for calibration and validation of satellite products; 5) complement, extend or potentially replace field surveys (especially in areas that are difficult to access); and 6) provide a scaling tool between field and satellite data.

Most importantly, this rapidly emerging technology provides an opportunity to explore new insights into hydrological, geomorphological, atmospheric, and biotic processes, and represents a real game-changing sensing platform.

In a recent contribution, Vivoni *et al.* (2014) reviewed the application of UAVs for ecohydrology and suggested that UAV remote sensing can fundamentally change how ecohydrologic science is conducted. This same is true for hydrology. At the

most basic level, UAVs can provide turn-key solutions of ultra-high resolution RGB imagery using consumer-grade cameras. Recently, multispectral and thermal sensors have gained traction and are increasingly being deployed by scientists. While lidar and hyperspectral sensors are still in an early operational phase, rapid progress is being made. One of the breakthrough technologies to the success of UAVs for mapping applications is structure-from-motion (SfM) and dense image matching (Turner et al., 2012). SfM is based on photogrammetric principles and generates detailed 3D point clouds

from overlapping and multi-view photography. UAV platforms are ideally suited to fly highly overlapping flight lines, and collect hundreds of overlapping images during dedicated campaigns. Extremely rich 3D information on the terrain, vegetation, buildings, geology, etc. can then be extracted cheaply and efficiently by the end-user. For hydrological applications, SfM provides information on micro-topography and can be used to generate digital terrain models (DTMs) and digital surface models (DSMs) at unprecedented detail. Apart from their natural affinity to application in precision

agriculture (Zhang and Kovacs, 2012) and vegetation health and stress monitoring (Zarco-Tejada et al., 2012; Zarco-Tejada et al., 2013), a number of recent contributions have demonstrated the utility of UAVs in hydrological process studies, with snow depth retrieval (Vander Jagt et al., 2015), flood mapping (Feng et al., 2015), irrigation monitoring (Bellvert et al., 2016) and evaporation estimation (Hoffmann et al., 2016) all being explored with these new platforms.

**Figure 3. Employing a UAV to retrieve high-resolution multispectral information on the land surface for hydrology and related applications over an Australian rangeland site located near Fowler's Gap in New South Wales. Retrieved products include: a) a false-colour infrared image; b) a reconstructed digital surface model using visible imagery and structure-from-motion techniques; and c) an optimized soil adjusted vegetation index (OSAVI) derived from the 4-band multispectral image. Images were captured using a MicaSense/Parrot Sequoia sensor on-board a 3DR Solo quadcopter. The UAV was flying at a height of 40 m, providing a**

**ground sampling distance of approximately 3 cm. Imagery provided by the University of Tasmania's TerraLuma Research Group.**





New UAV based sensor technologies are likely to drive further advances in hydrological process description and understanding. For example, advances in sensor manufacturing have now enabled production of frame-based hyperspectral snapshot systems that are much smaller than a typical consumer grade compact camera. Similar miniaturisation processes are

being applied to thermal sensors and laser scanners. These recent developments offer opportunities to the hydrological community by allowing the combination of multiple sensors that acquire data simultaneously. The acquisition of 3D information on terrain and vegetation, together with hyperspectral and thermal imagery, was previously a highly specialised task for very experienced airborne remote sensing crews. Now, this multi-sensor capability is rapidly becoming available for UAV platforms, providing unprecedented information for the remote sensing of the environment. However, as with any new

technology, UAV deployment comes with challenges as well as opportunities. One potential threat to the success of UAV remote sensing is that innovations are primarily driven from a technological rather than scientific perspective. While new airframes and sensors are evolving at an impressive pace, research is required to deliver rigorous processing workflows and to generate accurate and robust end-products that are meaningful. There is a real risk that new sensors and products may produce little more than "pretty pictures" without a thorough understanding of sensor performance, precision and calibration.

Semi-automated processing workflows are needed to ensure accurate geometric, radiometric, and spectral corrections. These workflows will have to cope with a data deluge of hundreds to thousands of images that typical flight campaigns generate, but developments in cloud-computing (Sect. 3.8) may provide a solution to the currently long processing times. Furthermore, as the need (or desire) for ultra-high resolution imagery increases, there will be a push to extend UAVs beyond visual line of sight (BVLOS) in order to cover larger areas. Visual line of sight is a current legal limitation of UAV operation in many

countries, which effectively limits the size of the study area to an order of 1 km$^2$, making the retrieval of information over larger catchments a laborious and time-consuming process. Improvements in technology and safety will ultimately make BVLOS operations feasible, but it will take time for regulatory bodies to keep pace with advances in technology.

Even though UAV remote sensing requires expertise in piloting, sensor operations, calibrations, and image processing

workflows, it is now possible for small groups and even individual end-users to collect their own ultra-high resolution multi-sensor EO data: a capability that even a decade ago, was the purview of space agencies and highly specialised airborne data providers. In the not too distant future, fully autonomous systems are anticipated. Although current applications are some way off being completely autonomous, the ultimate goal of the UAV is analogous to the image capturing capability of the space-based satellite: a self-propelling, powered, self-contained and independent data collection system. So long as the

needed developments in UAV science can keep pace with the rapid technological innovations, these innovative observation platforms are well placed to deliver needed advances in hydrological understanding.





### 3.3 Balloons and Solar Planes

UAVs are not the only non-orbiting remote sensing systems driving progress in hydrological observation: they are just one of the latest. Aerial weather balloons have been used for more than a century to remotely monitor terrestrial systems. Some of the earliest uses of balloons were to carry observers over battlefields throughout the 1800s and even during World War I,

providing an unparalleled logistical and military planning tool. Today, balloon designs enable a low cost, stable platform for intriguing hydrologic and related remote sensing applications (Chen and Vierling, 2006). Apart from providing soundings of atmospheric temperature, pressure, and humidity, along with a variety of other meteorological variables, a range of enhanced measurement capabilities are also possible. Vierling et al. (2006) constructing a tethered balloon consisting of meteorological instruments, GPS receiver, thermal infrared camera, and a video camera, all operating in real time with data

downlinked to a receiving computer. A more recent and novel application was the use of a mobile laser scanning lidar attached to a tethered balloon to acquire topographic elevation measurements (Brooks et al., 2013; Hauser et al., 2016). Costing approximately $100,000 the approach yielded a point-cloud of elevation measurements accurate to about 5 cm and spanning an approximately 75 m swath along the balloon's trajectory. With thousands of data points per square meter, the system offered sub-meter spatial resolution for accurate surface topography. Another system was developed by Shaw et al.

(2012), who retrofitted a tethered balloon with red and infrared imaging capabilities for less than $1000, providing an approximately 12 cm spatial resolution in a 64 m wide imaging swath from a legally restricted flying height of 50 m.

Like balloons, aircraft based remote sensing has existed since the earliest developments of powered flight. Since more traditional aerial methods are well-known and easily accessible via the peer-reviewed literature (Green et al., 1998), we

focus here instead on a more speculative but intriguing sensing future. Consider the recent around-the-world piloted flight of the Solar Explorer 2 (2016), an entirely solar powered airplane weighing 2,300 kg and having a 72 m wingspan. Covered in more than 17,000 photovoltaic solar cells, the craft achieved a maximum flight leg lasting almost five full days and nights. While this experimental system cost more than $200M, it highlights the future possibilities of having unmanned aircraft flying uninterrupted over fixed locations, without the need for landing. Back of the envelope calculations, assuming an

average velocity of 75 km/hr and a maximum piloted altitude of 8,500 m, suggest that a similar unmanned plane equipped with an imaging sensor capable of 20 km swath widths could observe areas of 300 km by 120 km in a single day: enough to sense the extent of the Sierra Nevada Mountains and their snow packs in about three days.

Unsurprisingly, improved Earth observation is not the only motivation driving the exploration of balloons and solar-powered

platforms. A number of Silicon Valley tech companies have well developed plans to use unmanned systems to deliver broadband internet coverage to poorly connected regions of the globe. Google's Project Loon (https://x.company/loon/) is perhaps the most advanced of these and is based on the idea of using stratospheric winds to navigate and control an interconnected network of high-altitude balloons. Using this approach, the project aims to provide internet access to both



developed and developing communities. With a similar goal in mind, Facebook's Project Aquila (Zuckerberg, 2016) is a parallel effort exploring solar powered aircraft. Aquila's aim is to have a fleet of planes flying at between 18,000–27,000 m that would stay aloft for months at a time, using on-board lasers to transmit and receive information to users below. A first unmanned flight was completed in late June 2016, lasting for 96 minutes (Gomez and Cox, 2016), but many technical

barriers remain to be overcome. While these examples are focused on providing communications infrastructure to the estimated 2 billion people currently without internet access (representing an untapped revenue stream relative to the largely saturated market in most developed countries), there are clear opportunities for leveraging such systems for enhanced EO. Harnessing a fleet of high-altitude aircraft with an array of lightweight sensor packages provides a platform not just for opportunistic sensing, but also for evaluating new technology, calibration and validation of satellite systems and supporting

large scale test-beds for product assessment: the last representing an often ignored (or under-funded) element of space based Earth observation. Leveraging the advances in technology behind the development and ultimate production of these systems may provide scientists with direct access to their own airborne platforms, offering capabilities to individuals or research teams that are currently beyond the scope or reach of most. While such future platforms remain somewhat speculative, these early developments are not just exciting, they represent real pathways towards an enhanced Earth observation strategy.

**3.4**     **The Rise of the Cubesat**

The demand for increased spatial and temporal resolution is one of the underlying drivers of sensor and platform development, with the assumption being that enhanced resolution will improve the monitoring, characterization and understanding of terrestrial ecosystems. Till recently, there has been a rather incremental improvement in observing system specifications. Current agency based high spatial resolution satellites, such as the Landsat series or Sentinel-2 platforms,

provide spatial detail at the 10-100 m resolution, but are constrained by the temporal frequency of acquisitions (10–16 days). When considering the influence of cloud cover on the visible, shortwave infrared and thermal infrared portions of the spectrum, data continuity and availability can be severely impeded (Roy et al., 2008). While deploying two identical sensor systems, as with Sentinel-2A and -2B (Drusch et al., 2012), represents significant progress towards improving the temporal resolution, acquisition of near-daily high-resolution imagery can only currently be met via expensive tasking of commercial

multi-sensor satellite systems such as RapidEye and WorldView (Houborg et al., 2015), and even then on an area-limited basis.

One way in which enhancements in revisit time and large area availability can be realised is via the launch of a larger number of replicate sensor systems. In the past, such attempts have been hindered by the high mission costs of the type of

large satellites favoured by space agency missions. For instance, Landsat-8 (which is around the size of a large car), had an estimated cost of $855M to build and launch, so producing multiple versions (and associated launch costs) is not a realistic proposition. The 2014-2020 budget for the European Copernicus Earth observation program, which includes the Sentinel missions, is estimated at approximately €4.3B, but does not currently include multi-satellite constellations beyond the

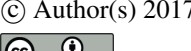


Sentinel-2 pair (Denis et al., 2016). The key limitation in the replication of multiple sensing platforms is related to the satellites size and the associated price tag. One possible solution to this constraint that has seen some impressive real-world results is an obvious one: make satellites smaller and lighter and they become cheaper to launch. Such an approach is behind the CubeSat concept, introduced by Stanford University and the California Polytechnic State University in 1999 (Puig-Suari

et al., 2001). CubeSats have provided the foundation upon which the recent surge in the development and launch of constellations of compact (i.e., 0.1 – 10 kg) pico- and nano-satellites (Bouwmeester and Guo, 2010; Selva and Krejci, 2012) can largely be attributed. A single-unit (1U) CubeSat, measuring 10 x 10 x 11.35 cm$^3$ and typically weighing less than 1.33 kg, forms the base level building block for a range of larger configurations. Indeed, CubeSats can be configured in a variety of sizes, increasing as integer multiples such as 3U, 6U or 12U, to expand observation capacities and potential applications

(Hevner et al., 2011). The advances driving CubeSats have not occurred in isolation, nor are they solely a product of economies of scale. The economics of space observation is changing rapidly, due to a combination of sensor miniaturization (allowing the development of standardized smaller satellites comprised of commercial off-the-shelf (COTS) components) and their deployment as secondary payload on commercial and public launch platforms (Woellert et al., 2011). The emergence of reusable rockets is also a major driver in the cost reduction of actually placing infrastructure in orbit, making

the launch of investigator led CubeSats a feasible proposition (see further details in Sect. 4.2).

Regardless of the driving forces behind their emergence, CubeSats represent a cost-effective observation strategy that provides a unique opportunity for the implementation and demonstration of technological innovations, serving as potential test beds for advanced sensing systems or even as direct replacements to larger satellite missions (see e.g. NASA's CubeSat

Launch Initiative; (NASA, 2016a) and Small Spacecraft Technology Program (NASA, 2016b)). NASA's Jet Propulsion Lab (JPL) is already exploring the CubeSat potential, with new on-board processing and sensor technology testing being conducted on planned CubeSat missions (Edberg et al., 2016). From a hydrological perspective, JPL's RainCube (Haddad et al., 2016) that is scheduled for launch in 2017, will act as a demonstration mission for the use of Ka-band radar for precipitation retrieval. Another JPL project is the CubeSat Infrared Atmospheric Sounder (CIRAS), that seeks to match some

of the temperature and water vapour profiling capabilities of the AIRS instrument (Aumann et al., 2003), but on a considerably smaller platform. Driving these efforts is the opportunity to leverage the significantly reduced cost, relative to conventional satellites, that makes launching constellations or swarms of CubeSats economically feasible. They also represent an inherent risk minimisation strategy: a systems failure on a sole-satellite configuration is mission ending, while multiple failures could occur within a constellation and still retain its mission capability. Such an approach has the potential

to revolutionize monitoring capacity from space, not just from a hydrological perspective, but across disciplines and sectors.

A number of commercial companies are leading the way in exploiting this observation strategy, most notably Planet (formerly known as Planet Labs; www.planet.com) who, with more than 150 3U CubeSats launched since 2013, manages the world's largest constellation of satellites in orbit (Planet Team, 2017). Planet's flock of "Doves" are capable of capturing





RGB and near-infrared imagery at 3-5 m ground sampling distance (GSD), providing near-daily global coverage based on a full constellation of nano-satellites. This emerging resource provides new and exciting opportunities for a wide range of applications seeking to exploit high-resolution clear sky imaging. One recent example using these data is the retrieval of high resolution Normalized Difference Vegetation Index (NDVI) for precision agriculture (Houborg and McCabe, 2016), but there are clear applications in land cover and land use change detection, environmental monitoring and numerous other fields of interest (see Fig. 4). The CubeSat approach features in other commercial enterprises, such as the planned Astro Digital Landmapper-HD constellation (scheduled for launch in 2017) that comprises twenty 6U CubeSats capturing five spectral bands at a GSD of 2.5 m every 3-4 days (www.astrodigital.com). Planetary Resources (www.planetaryresources.com) envisions a programmable constellation of ten 12U CubeSats, delivering visible to near-infrared (400-900 nm) hyper-spectral and mid-wave (3-5 μm) thermal infrared data at 10-15 m GSD for any spot on Earth on a weekly basis. With the cost of a CubeSat ranging anywhere from a few tens of thousands upwards (including launch costs), the prospect of investigator or community driven missions becomes a realistic proposition.

**Figure 4. Multi-scale capabilities of state of the art sensing optical satellites. Image illustrates the expanding resolution options available from both commercial and government satellites. A) Planet CubeSat at 3 m ground sampling distance over the Tawdeehiya Farm in Al Kharj, Saudi Arabia. Center pivot irrigated fields dot the landscape, with dimensions approaching 800 m. The inset in A) is zoomed to show the resolution advantages offered by the next generation of sensing solutions over B) Landsat-8 at 30 m, with C) Sentinel-2A at 10 m and D) Planet imagery at 3 m providing enhanced details. All images are false colour representations of NIR, Red and Blue in RGB bands. Sentinel-2A and Landsat-8 images were acquired on December 4th, 2016, while the Planet data were captured on December 5th, 2016.**

Instead of launching constellations (i.e. a large number) of independent satellites into space, others have advocated the concept of a dense network of distributed space missions working in cooperation, where sensing systems coordinate to achieve a monitoring task in much the same way as a distributed sensor network collects information on the ground (Barnhart et al., 2009). Using satellite-on-a-chip or printed circuit board approaches, such low-cost, sub-kilogram options have obvious potential for hydrological and related sensing. While the next generation of CubeSats has the potential to revolutionize Earth observation, data from such platforms should ideally complement, and not necessarily replace, the high quality imagery that is currently acquired by conventional large satellite missions. To harness the potential and exploit these technological advances demands preparation (Dash and Ogutu, 2016) and this will only be realised through synergistic exploration and leadership from government space agencies, the science community, and increasingly the private sector. An underlying assumption here is that space junk will not continue to accumulate to the point of becoming an intolerable risk to launching satellites to low Earth and geosynchronous orbits: though that dystopia would actually enhance the importance of the sub-orbital, alternative technologies described throughout this section. Whether intrinsic barriers (e.g. payload launch) or a divergence of commercial motivation versus scientific research interests will inhibit this exciting and much needed development in EO are topics that are explored further in Sect. 4.2.





### 3.5     Mobile Phones and Citizen Science

While space-based and near-Earth sensing platforms are revealing entirely new avenues of EO, there are technologies closer to home that are also revolutionising how we can monitor, sense and interact with the environment. Smartphones have transformed entire societies, from the most developed countries to regions where a regular source of electricity or freshwater is still lacking. In Africa, mobile banking has allowed Kenya to lead the world in mobile money via its M-PESA system (Economist, 2013). Indeed, sub-Saharan Africa is one of the world's fastest growing regions for mobile phone subscribers, numbering more than 330 million as of mid-2016[2]. As such, smartphones present as a ubiquitous platform from which to harness the possibility of remote sensing of hydrologic variables. For example, both river discharge and rainfall measurements are key requirements for constructing an accurate water balance. As is the case for many regions in the world, archived records of these measurements are available across the entire Congo River Basin from the 1930s to the 1950s. Unfortunately, the hardcopies containing these records sit on shelves in Kinshasa and Kisangani. In the most simple application of mobile technology, these records could be photographed, digitized and uploaded to the internet using a smartphone. While such efforts would be tedious, these data represent invaluable climate records for the world's second largest river basin. In this sense, a person with a smartphone becomes a remote, or at least proximal, sensing platform capable of providing much needed hydrologic information.

However, such a basic application of smartphone technology belies the potential these devices have in providing a distributed measurement network. Plug-in and Bluetooth technologies linked to smartphones enable potentially billions of users to become mobile platforms for measuring actual hydrological events. Data from 2013 estimated that there are 7.3 billion mobile subscriptions globally, with 3.2 billion of these linked to smartphones[3] (see Fig. 5). Undoubtedly this number has increased in the last few years. As an example of the immediate potential of this sensing platform, iBobber is a $100 baseball-sized fishing bob that measures water depths and temperatures and has GPS location capabilities (see http://reelsonar.com; noting that there are other similar devices available on the market). Fishermen everywhere could be recording water depths for river hydraulic models and for total storage in lakes. In a more focused manner, teams of lay-scientists could be easily trained to use such low-cost devices to provide remotely sensed water depths in cost effective ways, e.g., a leisurely riverboat excursion or simple fishing pole cast from the shoreline to yield water depths. It is not hard to envisage numerous other smartphone-enabled devices that auto-upload their measurements to the internet. Indeed, it is the ubiquity of smartphones that enables the imagining of new hydrologic measurements.

**Figure 5. Worldwide global system for mobile communication (GSM) coverage for the year 2013. The GSM network does not include the growth of related 3G or 4G networks. The image is derived from Figure 2 in Overeem et al. (2016).**

---

[2] http://interactive.aljazeera.com/aje/2016/connecting-africa-mobile-internet-solar/connecting-africa.html
[3] https://www.ericsson.com/mobility-report




However, there are (at least) two challenges with such citizen science: 1) making certain that the measurements are accurate; and 2) connecting the hydrologic researcher with the smartphone users. Both challenges are solvable using standard methods employed in hydrological sciences. For instance, data assimilation and other statistical approaches can ensure that

measurements collected from disparate platforms are appropriately integrated in hydrologic models. In terms of engagement or outreach, cross-disciplinary interaction between the social and physical sciences could facilitate the implementation of strategies to effectively engage citizen science. One application where smartphones have already demonstrated their potential for environmental monitoring is their use as thermometers. Overeem et al. (2013b) showed that thousands of smartphone battery temperatures uploaded to a central database through an Android application could be employed to estimate daily

mean air temperatures in eight major cities around the world with reasonable accuracy. Their results show the potential of crowdsourcing for real-time temperature monitoring in urban areas, where dedicated temperature measurements by meteorological services are typically scarce. Recent reviews have further illustrated the success of a number of crowd-sourcing projects, detailing the use of mobile video and imagery to capture and analyse flash-flooding, debris flow and flow velocities (Le Coz et al., 2016), precipitation events (Allamano et al., 2015) as well their application in atmospheric and

climate sciences (Muller et al., 2015), detailing an exciting avenue of enhanced data collection.

Importantly, crowdsourcing in hydrology is not solely about smartphones. De Vos et al. (2016) report on an effort to "crowd-source" rainfall data from personal weather stations in Amsterdam, exploiting the proliferation of low cost stations designed for home-based meteorological collection. Even in this single-city focused example, more than 60 inhabitants were found to

operate personal weather stations equipped with simple tipping bucket rain gauges within the Amsterdam metropolitan area, significantly increasing the sole rain gauge operated by the Royal Netherlands Meteorological Institute (KNMI) at Amsterdam's Schiphol Airport. While there are undoubtedly issues associated with poor siting considerations, (lack of) maintenance and (interrupted) connectivity that would need to be accounted for, the utility of such additional hydrological monitoring is obvious. Indeed, the De Vos et al. study highlighted the additional information on the space-time variability of

rainfall over a densely populated area that could be retrieved with reasonable accuracy and reliability from such a citizen network. In a particularly novel example of exploiting existing networks of data, Rabiei et al. (2016) inferred rainfall by utilising a vehicles GPS location together with sensors attached to the cars windscreen wipers. Many late-model vehicles employ infrared (or optical) sensors to determine rainfall intensity in order to automatically adjust the wiper-rate, offering the possibility of providing distributed records of rainfall: albeit limited to the road-network.

The use of non-traditional sources of information to infer, improve or inform upon our hydrological understanding, as well as to expand the distribution and spatio-temporal representation of existing networks, is a rapidly growing field that presents clear potential. The topic is explored in the section below, which details further related examples of opportunistic sensing.





### 3.6    Signals of Opportunity

For decades, hydrologists have relied upon dedicated ground based measurement equipment to deliver their observational needs. Such instruments are typically owned and operated by government agencies and regional or local authorities. Installed and maintained according to national or international standards, they offer accurate and reliable point observations of state
variables and fluxes. One drawback of these operational measurement networks is that they are spatially discontinuous, often lacking the required spatial and/or temporal coverage required for high-resolution real-time monitoring or short-term forecasting of rapidly evolving hydrological systems. Another drawback is that such networks are often costly to install and maintain, which makes it a challenge for nations in the developing world to operate them on a continuing basis. As mentioned in Sect. 2.3, conventional observations networks have been in a state of decline for a number of decades. Yet, our
modern world is full of sensors, from the cars we drive, to the mobile phones (and cameras) we carry in our pockets. We are in the age of the Internet of Things (IoT), where every day physical devices are connected to the network, sensing the world around us. Although related to the concept of "citizen science" that was introduced in Sect. 3.5, we couch the present discussion under the context of "opportunistic sensing": the concept of utilising signals from often unrelated measurements to inform upon hydrological processes. Inferring hydrological properties by making use of signals of opportunity is a
growing area of research. Essentially, it involves measuring how radiation from a source designed for one purpose changes as it passes through the atmosphere or reflects off of a medium of interest, which can then be interpreted to describe an often unrelated process. Such sources include radio, television, and other communications transmitters.

Telecommunication engineers have known for a long time that radio signals propagating from the transmitting to receiving
antennas of microwave links used in cellular communication networks are attenuated by rainfall. By using this knowledge, researchers have been able to translate this electromagnetic "noise" into a hydrometeorological "signal" (Messer et al., 2006; Leijnse et al., 2007). Indeed, it turns out that for the radio frequencies typically employed in such cellular networks, the signal attenuation is nearly linearly related to the average rainfall intensity. The attenuation can be inferred from the transmitted and received signal levels, which are operationally stored by telecommunication companies at regular time
intervals (typically 15 minutes or less) to monitor network quality. As these links typically have lengths of a few kilometres and are installed at just a few tens of metres above the ground, they can be considered as path-averaged rain gauges, well suited for hydrological applications. Several thousand such links covering the surface area of an entire country have recently been used to produce 15-minute rainfall maps of comparable quality to those obtained from gauge-corrected ground-based weather radars (Overeem et al., 2013a; Overeem et al., 2016). In addition to rainfall monitoring over urban areas (where
network densities are generally high), this technique offers much potential for high-resolution measurement in areas where the density of ground-based monitoring networks (i.e. gauges or radars) is typically low, such as in developing countries (Doumounia et al., 2014; Gosset et al., 2016). Given their spatial and temporal advantage, a number of research efforts have also sought to exploit geostationary based satellite platforms to infer hydrological processes. Indeed, there is a long history

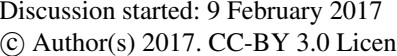



of using radio occultation measurements via the Global Positioning System (GPS) of satellites to infer atmospheric variables and profiles (Kursinski et al., 1997) of use for numerical weather prediction. More recent work has sought to expand the type of measurements that can be inferred between satellite links and ground stations. For example, Barthès and Mallet (2013) describe the use of an Earth-space link in the Ku band to measure rainfall, leveraging several hundred geostationary

5 telecommunications satellites transmitting in this frequency to infer periods of rainfall via signal propagation through the troposphere. Such information is not only useful for hydrological applications, but also for ground validation of satellite-based rainfall retrievals.

While improved representation of rainfall is of great importance to hydrological studies, soil moisture plays an equally

10 significant role in many process investigations. One of the best example of opportunistic sensing to date is the Plate Boundary Observatory (PBO) $H_2O$ initiative, which uses reflected GPS signals to estimate soil moisture (Larson et al., 2008), snow depth (Larson et al., 2009), and vegetation growth (Small et al., 2010). Advantages of this technique are that it provides temporally continuous data at scales ($\sim$1,000 m$^2$) between point measurements and satellite remote sensing footprints, and that cloud cover and labour are not issues. The use of proximal remote sensing techniques to measure soil

15 water content and soil properties at depths deeper than remote sensing capabilities (i.e. greater than 5 cm) represents an area of considerable importance in the hydrologic sciences. Another approach that aims to fill this observation gap is the COsmic-ray Soil Moisture Observing System (COSMOS) (Zreda et al., 2012), which provides an increasingly rich dataset for calibration, validation, and evaluation of remote sensing products and land surface models. Comprising a network of more than 200 cosmic-ray neutron probes at fixed installations across six continents, these data represent a valuable source of

20 independent information from which a range of hydrological responses may be inferred or assessed (Jana et al., 2016; Montzka et al., 2017). A recent addition to the COSMOS program has been the use of mobile "rovers", which offer a way to increase the spatial coverage from the local to mesoscales (Desilets et al., 2010; Chrisman and Zreda, 2013), while also offering a means to merge data from fixed probes to provide a multi-scale real-time soil moisture product (Franz et al., 2015).

In addition to supporting hyper-resolution land surface modelling needs, the rover approach provides opportunities not just in research, but also commercial activities: most notably in precision agriculture. For example, mounting rovers to existing farm equipment (sprayers, tractors, etc.), autonomous farm vehicles, or to rotating infrastructure (i.e. centre-pivot irrigation systems) offers an interesting opportunistic sensing possibility. The piggy-backing of data collection with existing farm

30 operations may provide a cost effective strategy to help inform the timing and spatial distribution of a range of on-farm information needs, from irrigation depths and optimal fertilizer application, to seed planting density and timing. The capacity to mount probes on delivery trucks, self-driving vehicles or even national train networks, would further expand observational capacity and provide semi-repeatable local and regional mapping opportunities across both natural and urban landscapes. Mobile sensors can easily collect data from either ground vehicles (e.g. snow mobiles, dog sleds, etc.) or low flying aircraft,





which offers a potentially unprecedented calibration, validation, and evaluation dataset for a range of hydrological variables. While roving probes are fairly heavy (50+ kg) and miniaturization options are somewhat limited, the use of drone swarms with several smaller probes functioning as a single unit would further increase mapping possibilities (see Sect. 3.2). With the simultaneous use of several detector energies (bare, cadmium shielded, and plastic shielded probes) recent research has

illustrated the means to collect information on vegetation condition, soil organic properties, and soil moisture simultaneously, providing a valuable resource to support observation and modelling strategies (Andreasen et al., 2016). Such sensing technology has the potential to augment on-going digital soil mapping efforts underway across the globe (Sanchez et al., 2009), as well as aid in the validation of existing high resolution products (Chaney et al., 2016).

In a final example of opportunistic sensing, we explore the potential of commercial passenger and cargo aircraft as mobile airborne sensing platforms. While observations from dedicated aircraft are typically only collected during sensor testing and infrequent, targeted measurement campaigns, there is little to inhibit (at least from a scientific perspective) airborne sensors from hitching rides aboard commercial aircraft, which would greatly expand their spatial and temporal coverage. Many airliners are already outfitted with Doppler radar and Aircraft Meteorological Data Relay (AMDAR), which provides

measurements of meteorological variables that include temperature, wind vector, and dew point temperature, and are made available for assimilation into weather forecast models and for scientific investigations deemed beneficial to the airlines. Advanced sensors for measuring water vapour more precisely have also been tested alongside AMDAR sensors. However, while such an approach to remote sensing has been espoused for more than two decades (Fleming, 1996), it has not as yet been employed routinely to enhance hydrological observation.

**3.7    High-definition Video from Space**

One of the most exciting remote sensing opportunities that has the potential to change not just the way we observe the Earth system, but also the manner in which we can use data to inform on processes, is the emergence of high-definition (HD) digital video. This game-changing visualization approach builds on a surprisingly long history of employing airborne video in EO studies (King, 1995). Indeed, some of the earliest satellite missions such as Landsat 1-3 (Townshend, 1981), used

vidicons, a type of cathode ray tube employed in capturing television images (Nagy and Nagy, 1972). These early instruments were basically 2D image snapshots, due to the temporal sampling limitations of the deployed systems i.e. essentially television cameras providing still photographs (Vaughan and Johnson, 1994). While the use of airborne and ground-based optical and multispectral video systems has been explored actively in vegetation and agricultural studies (Everitt et al., 1991), it is only in very recent times that the capacity to exploit full-motion HD video from space has

emerged. Indeed, it is this opportunity to utilise the temporal insights that HD video allows that represents the truly revolutionary aspect of this observing system.



With full-motion video imagery comes the capacity to capture dynamic meteorology, such as real-time cloud formation and dispersal or storm development and release: insights that could change our fundamental understanding of these processes. An ability to record the Earth system in real-time on a repeatable basis has inter-disciplinary implications. Pollution monitoring, disaster management and response, ecosystem assessment, as well as numerous and immediate hydrological

applications are imaginable e.g. flow velocity, flood propagation, erosion monitoring, contaminant transport and dispersion, precipitation tracking, to name but a few. One novel application lies in the use of satellite video data to reconstruct a digital surface model (d'Angelo et al., 2016) via structure-from-motion type approaches, providing insights into landscape changes in ways that static elevation datasets cannot. Being able to record debris flow down a river, or dynamic inundation in natural or urban systems could provide new insights into how we model, forecast and predict flow and related hydrological events.

However, while the possibilities of video imagery from space are exciting, as a discipline we are under-prepared to utilise such data effectively. Ultra-high temporal resolution information is not something that we routinely deal with, so how to exploit such data will require innovation and imagination. An obvious constraint in current modelling application is that the temporal resolution of even the most advanced hydrological schemes are usually on the order of minutes rather than seconds (Berne et al., 2004; Ochoa-Rodriguez et al., 2015). Direct ingestion is the most obvious (but least imaginative) manner in

which video data could be used, but computational and model-physical constraints are apparent. So, while a range of applications can be imagined, the practicalities of integrating or ingesting high-temporal sequences into our current modelling or analysis frameworks remain largely unexplored. Indeed, video imaging and analysis is more the domain of the computational scientist than the hydrologist, so these disciplinary lines will need to be crossed to take advantage of such technological breakthroughs. Although the potential applications are many, a paradigm shift away from the use of periodic

2D snapshots will be required to exploit the feature rich temporal dimensions offered by video streams.

It is important to note that this is not blue-sky research: the technology exists, satellites are already in orbit and data streams are available: but we are not keeping pace with the rapid advance in imagery possibilities. Indeed, it is the private sector that is leading the charge in realising and utilising the technology, with Google's SkyBox (now TerraBella;

https://terrabella.google.com/) providing high spatial (approx. 1 m) and temporal (30 frames per second, fps) full motion video imagery (Murthy et al., 2014). UrtheCast (https://www.urthecast.com/) is another company exploring this potential, with similar spatial (1 m) but lower temporal (3 fps) specifications (see Multimedia 1): although the second generation UrtheCast system that is due for launch in early 2017 will provide imagery at 0.5 m and 30 fps, in addition to a 1 m X- and 5 m resolution L-band synthetic aperture radar (Beckett, 2015). At the moment, both video platforms are limited to between

60-90 second captures, but expanding this technology to allow full-coverage real-time observation in low Earth orbit has been proposed on micro- and nano-type satellite configurations (Han et al., 2015), while others have presented a vision of a geostationary space surveillance system (Airbus GO-3S) (Villien et al., 2014). It is this combination of high-spatial and high-temporal observation that has the potential to dramatically alter the very nature of Earth observation.



**Multimedia 1. On-board the International Space Station, the UrtheCast IRIS high-resolution camera (HRC) capture colour video at 3 frames per second for a duration of 60 seconds. Here we see an example of the HD Video over the Burj Khalifi in Dubai. The tracking of vehicles on roads is analogous to monitoring flow in rivers or the speed of moving clouds, while the capacity to extract 3D structure of the underlying terrain provides opportunities in dynamic monitoring of surfaces.**

### 3.8    Cloud Computing and Data Analytics

While the capacities of today's EO sensors to collect data of relevance to hydrology are truly unprecedented, the challenges faced when trying to turn the raw satellite data into useful and trustworthy information can be daunting. It is by no means clear when and to what extent hydrology will start benefitting from the much increased volume and diversity of EO data.

However, the speed of adoption will likely be determined by the time it will take to move the EO data and their processing into the "cloud". At a very basic level a cloud can be understood to be a large-scale computing infrastructure capable of delivering services over the internet. A key enabler of cloud computing was the construction and operation of extremely large-scale, commodity-computer data centres at low-cost locations to achieve economies of scale (Armbrust et al., 2010). Nowadays, with falling prices for storage and computing, thematic aspects and service quality is becoming more and more

important. The advantages of cloud computing include, amongst other, virtualized resources, parallel processing, and data service integration with scalable data storage (Hashem et al., 2015).

In parallel to developments seen in other fields, novel EO satellites are acquiring data at a staggering rate, where even a single satellites collection can exceed many terabytes on a daily basis. Hence, over the regular lifetime of a satellite, more

than a petabyte of raw satellite data can easily accumulate. Clearly, processing such large data volumes is impossible with standard computing resources, nor is it meaningful to distribute the data over the internet, thereby replicating many thousands of queries. Instead, the only way forward will be to "bring the users to the data". In practical terms, this means that EO data processing will increasingly take place in large virtualised data centres, i.e. in the cloud, allowing large numbers of users to access the data and enabling collaboration on the development and use of EO services. With such infrastructure, it

then becomes possible to start building multi-level EO data processing chains in a collaborative manner. Ideally, interpretive models and subsequent data analysis would also be run where the EO data reside, ensuring a seamless processing line from the raw sensor data to the final hydrologic predictions, allowing each expert along the value-adding chain to focus on his or her competencies. Considering the increasing complexity of scientific algorithms and models used in EO and hydrology, such collaboration can be expected to speed up research and development efforts, leading to a much faster data uptake in

hydrological practice.

The adoption of cloud computing technologies in EO and hydrology will not be without its challenges. Apart from the practical software-based considerations that allow virtualisation of large computing infrastructures with hundreds to thousands of users, a much larger obstacle is how best to organise the expert community, ensuring that joint efforts to





develop code and products lead to quality controlled, well documented, and user friendly software and data. One of the most advanced cloud platforms is Google's Earth Engine (http://earthengine.google.com), which provides a platform for petabyte-scale scientific analysis and visualization of geospatial datasets, both for public benefit (non-commercial use is for free) and for business and government users. Its data catalogue contains a wide variety of popular, curated datasets, including the

world's largest online collection of Landsat scenes (Gorelick, 2013). Amazon Web Services offers a similar storage and analytics platform, which houses an expanding collection of satellite, meteorological and climate datasets available to the user community, including recent Sentinel-2 data and a number of NASA collections (aws.amazon.com/earth).

A number of early examples have explored the hydrology related opportunities afforded by cloud-based platforms (McGuire

et al., 2014; Astsatryan et al., 2016). Donchyts et al. (2016) employed the Earth Engine for mapping surface water changes at 30 m over the past 30 years on a global scale, an effort that would not have been possible without such data analytic centralization. While the Earth Engine is popular amongst scientists, Amazon's cloud is increasingly being used by commercial companies to showcase their EO services, such as the Sentinel-2 web mapping service offered by Sinergise (www.sentinel-hub.com). Another cloud platform serving both EO and hydrological applications is currently being built by

the Earth Observation Data Centre (EODC) for Water Resources Monitoring (https://www.eodc.eu/), a public-private partnership with a goal to foster the use of EO data for monitoring global water resources (Wagner et al., 2014). In addition to optical data (i.e., Landsat, Sentinel-2) EODC holds a complete global archive of Sentinel-1 Synthetic Aperture Radar (SAR), which can be processed with a supercomputer for continental to global scale mapping of soil moisture, water bodies and other hydrological parameters (Elefante et al., 2016).

Clearly there are many potential and diverse applications of cloud computing in hydrology, many of which are being enabled by access to the underlying Applications Program Interface (API), a common feature of many of the Silicon Valley type start-ups. Although representing rather focused examples of cloud computing opportunities, the cases noted above serve to illustrate that this revolutionary change in technology, which has the potential to completely overhaul working practices in

EO and hydrology, has already started. As the spatial and temporal resolution of EO data increases, the development of efficient cloud computing, storage and on-the-fly processing solutions becomes even more relevant. This is especially pertinent for a community that seeks to embrace the concept of hyper-resolution hydrological modelling, where the scales of processing and data requirements start to push-back on available computational power and resources (Bierkens et al., 2015). Undoubtedly, any future EO strategy in the hydrological sciences will have cloud computing as a core element, so

recognising and resolving the inevitable challenges and opportunities that cloud computing will bring to the community will be key to realising its potential.

A parallel consideration that will follow any increase in data volumes and the associated computing demands is the need to explore more efficient approaches to exploit and interpret the petabytes of satellite data being collected on a routine basis





(Warren et al., 2015). The era of big-data and artificial intelligence is upon us, whether as a community we are prepared for it or not. Traditional modelling and analysis techniques are ill-designed to interrogate or utilise such massive datasets, and alternatives based on machine-learning and data-analytic methods that can be used for regression or classification problems involving massively multivariate systems are becoming popular. These techniques have the potential to either completely
replace process-based models, or work in combination to make them less computationally expensive (Lary et al., 2016). Commonly used machine-learning methods include artificial neural networks, support vector machines, genetic programming, decision tress or random forests, amongst many other approaches. These methods are usually applied in a 'supervised' context, in which a database subset is used to train the algorithm to reproduce an expected response (i.e. 'learning process'), and a different subset can be used to test or validate the performance of the trained algorithm. An
interesting characteristic of these methods is that little to no knowledge of the physical processes underlying the observed variables is required to implement them, which releases their potential for discovering unexpected relationships as new hydrological and climatic observations become available (Faghmous and Kumar, 2014; Lary et al., 2016).

Machine-learning methods have been applied across a range of science and engineering applications for over two decades. A
number of recent examples have targeted the (retrospective) prediction or retrieval of hydrological states and fluxes from single- and multi-satellite sources, including the estimation of typhoon rainfall over the ocean (Chen et al., 2011), the retrieval of surface soil moisture (Rodríguez-Fernández et al., 2016) and water vapor content (Aires et al., 2001), the estimation of river runoff (Rasouli et al., 2012; Deo and Şahin, 2016), the analysis of global hydro-climatic controls on vegetation (Papagiannopoulou et al., 2016), the training of high-resolution sensors for retrieval of NDVI (Houborg and
McCabe, 2016), and the derivation of continental water and carbon fluxes using decision trees (Jung et al., 2009). Still, the application of these techniques to dynamically monitor hydrological events and processes using remote sensing is an emerging field, with relatively limited existing applications. With the storage and analysis opportunities afforded by cloud computing, the capacity to streamline many of these examples into on-the-fly applications is more a reality than ever before, providing a new and on demand observation and analysis source.

Despite this remarkable confluence of data science and remote sensing, one may still resist the narrative that there is no problem that a sufficiently complex machine-learning algorithm cannot unravel given enough data (Anderson, 2008). If this were the case, there would be no need for domain expertise to understand current and future challenges in hydrology: the dilettante will have prevailed (Klemeš, 1986). Indeed, there remain several obstacles to any predicted ascension of a
completely data-driven approach to hydrology. Observations of the hydrosphere often have a spatio-temporal structure that emerges in the form of correlations between variables, but this correlation may not necessarily imply causality. Therefore, being able to draw strong deterministic conclusions about the behavior of hydrological systems based on data-driven methods often requires prior knowledge (and understanding) of the physical processes (Faghmous and Kumar, 2014). As an example, Papagiannopoulou et al. (2016) discuss how the application of random forest models to auto-correlated vegetation

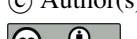

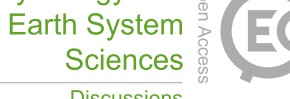

imagery and cross-correlated temperature and precipitation can lead to the wrong conclusion that temperature controls vegetation growth in water-limited regions. Changing sensors or satellites (e.g. as part of data continuity missions) can result in temporal gaps, discontinuities, and artifacts, as well as including sensor degradations, which without context, would impact the conclusions that data-driven models may yield on the behavior of hydrological systems. All of this is to say that

without subject knowledge, such temporal record adjustments are unlikely to be diagnosed or interpreted appropriately.

On the other hand, a dogmatic approach to a purely physically based hydrological process representation has inevitable limits to advancing understanding. The concept of "letting the data speak for itself" is particularly attractive in a discipline where so much of our physical understanding is based on a relatively simplistic description of process and function, and

where its application is routinely extended beyond the scales at which it was observed to be relevant. As both hydrological and remote sensing research progress, it is perhaps prudent that we seek the middle-ground, where the development of machine-learning methods might be guided by theoretical constraints and understanding, and that they be used to complement or improve more traditional physically-based models, which in turn can add interpretability in regards to the underlying processes. Regardless, the opportunities being presented by these new and innovative approaches are likely to

challenge our concept of hydrology as a discipline, especially as the exploration of inter-disciplinary datasets provide new insights and understanding to hydrological processes and behavior: a topic that is expanded upon in the context of a 4[th] Paradigm in Hydrology, as discussed in Peters-Lidard et al. (2017) (this issue).

## 4        The Changing Earth Observation Landscape

### 4.1      The Space Agency Approach

Space agencies are government entities that are tasked with undertaking and enabling the development of space based science and technology. In the United States, approximately 25% of NASA's $19B budget goes to funding the Science program, of which $2.0B is allocated to Earth Science[4]. With these resources, NASA supports 60 operating satellite missions, 35 that are in the planning stages, and over 10,000 U.S. scientists, including the award of more than 3,000 research grants (n.b. these include awards to planetary science, astrophysics, and Earth science). Other space agencies are smaller, but

still have $2-5B budgets e.g., ESA, ROSCOSMOS, CNES, DLR, and JAXA. While the budget numbers seem quite large, space agencies are still challenged to afford the suite of desired satellite missions that satisfy a diverse scientific community as well as government needs. The cost of design, launch and operation of a satellite mission has increased considerably over the last few decades. Satellite missions twenty years ago cost on the order of $100M, but today, they can reach up to (and beyond) $1B. Agency budgets, however, have not grown by a similar magnitude. Indeed, measured in 2014 dollars, NASA's

budget has remained around $20B for over three decades.

---

[4] https://www.nasa.gov/sites/default/files/atoms/files/fy_2017_nasa_agency_fact_sheet.pdf





To forecast the types of future missions that will be launched by space agencies, we can look to their planning process to evaluate the historical success at following such plans. The best known amongst the space agency planning efforts for Earth observation is the National Academies Earth Science and Applications from Space "Decadal Survey" (2007): an Herculean

effort that energised the Earth science community to gather and prioritize NASA's future EO capacity. The endeavour identified 15 new missions for launch as well as urging NASA to launch two additional missions already in mature planning stages, i.e. GPM and a replacement for Landsat 7. The GPM core observatory launched in February 2014, following Landsat 8 in February 2013. Of the original 15 new missions proposed in the Decadal Survey, SMAP (Entekhabi et al., 2010) is the only one to have launched (in January 2015). Other missions were already in various stages of planning before the Decadal

Survey, including the SWOT mission (Biancamaria et al., 2011), which was initiated five years prior to 2007. All of this is to illustrate that it is not unusual for government space agency missions to take on the order of two decades to go from concept to launch (see Sect. 3.1), and that the systems that move from proposal to orbit are not always identified by consensus. Indeed, sometimes an entire generation of scientists move through the community before the space-based measurement system arrives in orbit: certainly not a shining example of a fast-paced discovery process.

An important consideration, particularly in light of the "fast and nimble" approach advocated by Silicon-Valley driven commercial enterprises, is that by the time any government satellite actually reaches orbit, the technology on-board may already be a decade (or more) old. The obvious implication of this is that space agencies may not be launching the most cutting edge sensing platforms. Indeed, by their nature, space agencies are risk averse, seeking out the most robust

technology to survive the hazards of space and ensure delivery of mission objectives. This model stands in contrast to the technological advances being made today, especially in instrument design and function, which occur at a seemingly faster pace than in decades past. The emerging concept of "agile aerospace" combined with the opportunities being presented by commercial ventures via the rise of the CubeSat (see Sect. 3.4) and other sensing platforms, present an ideal test-bed for new technology and demonstrator systems: a theme that will be explored in the following section.

**4.2    The Commercialization of Space**

The commercial sector presents something of a counter example to the government space agency approach. Undoubtedly, commercial enterprises build upon the successes (and sometimes direct funding) of the government sector. However, recent advances have seen an increased capacity to combine that foundation with venture capital and new technology to provide immediate EO platforms to the paying customer. Of the recent players operating in this market, perhaps the most well-

known is Space Exploration Technologies (SpaceX) (www.spacex.com). Employing techniques such as 3-D printing to create strong and durable rocket parts at a fraction of the time taken for traditional casting, they have also reimagined and reengineered the reusable launch vehicle concept, representing a major innovation and cost-saving to the delivery of





payloads into space. An objective of these new rocket companies is to radically improve the efficiencies of payload delivery at a fraction of current costs, which have been estimate at up to $20,000 per kg (Coopersmith, 2011). Indeed, the SpaceX approach purports to reduce costs by about half compared to traditional launch vehicles (e.g., $62M for a 22,000 kg payload on a Falcon 9 rocket)[5]. With a launch date for early 2017, the SpaceX Falcon Heavy aims to reduce this cost further, lifting

up to 54,000 kg to low Earth orbit for $90M, or $1,700/kg (n.b. finding precise figures for this is difficult, as they are "reusable rockets" and the costs decrease as function of the number of planned launches). Rocket Lab (www.rocketlabusa.com), a New Zealand start-up, is offering smaller launch vehicle capability, but with greater frequency and selective orbit. Aimed specifically at the small satellite market, it will launch a 150 kg payload for $5M and also provide a ride-sharing option where users can launch 1U to 12U CubeSats, opening up the prospect of investigator led space

missions.

But getting to space is only one aspect of the recent rise in commercial activity. As discussed in Sect. 3.4, there are a number of companies exploiting technological advances in sensor miniaturization, reduced power consumption and improved battery life (that have been driven in large part by the mobile phone industry) to produce cheaper, smaller and more efficient satellite

platforms. One of the most ambitious of these ventures may be Planet (www.planet.com), a $200M five-year-old start-up with a stated goal of providing complete global coverage of the terrestrial surfaces of the Earth every day via a constellation of their CubeSat "Doves", representing an unprecedented high-resolution information resource (Houborg and McCabe, 2016). But Planet is just one of a number of non-agency based companies playing a role in EO: DigitalGlobe, Blacksky, Planetary Resources, Spire and TerraBella are just a few examples of private ventures that are operating largely independent

of government space agencies.

Apart from the motivation and rationale of these companies shifting towards profit making enterprises rather than operating for the social good, a key difference between government and commercial sector approaches to space is funding for scientific use. By very approximate calculation, NASA provides about one-tenth of a satellite missions cost for scientific

users. Thus, a $1B mission might provide on the order of $100M for related scientific activities. A private company, with a total budget on the order of a few hundred million dollars, would obviously place a much lower (or no) priority on directly funding the science community. However, while space agencies are certainly well motivated by science, the significant imbalance between technology and science funding indicates a strong vested interest in their supported technology engineering communities. In contrast, commercial enterprises are strongly motivated by profit i.e. venture capitalists expect

a return on their investment, so optimising efficiencies in production, launch and operation are paramount.

---

[5] See http://www.spacex.com/about/capabilities




There are numerous examples of private-public partnerships that have shown the success of industry engagement, and many opportunities exist to exploit intersections of interest not just within industry, but also with other government departments (e.g. Defence). Of course, putting satellites into orbit is only one small part of a space agency's mission. But what is becoming clear is that there are cheaper, faster and more functional options being presented to the community from a variety

of sources, both private and commercial, that present an opportunity to embrace a new era of EO beyond the traditional agency approach. In some ways, government space agencies are already adapting to leverage these changes in their own operations by sub-contracting out certain mission elements to the commercial sector e.g. resupply of the ISS using SpaceX Falcon 9 rockets, along with the many satellite components built by private companies under government contract. Still, it remains unclear how individual investigators can best leverage these new observational platforms and the data they produce

within the current mode of open-access, peer-review and publication of results. Will hydrologists be able to afford this data, and once provided, will there be limitations on its use? There is a very real risk that the successful commercialization of space could pose a serious threat to the function and operation of both space agency and investigator led Earth observation, as well as scientific advancement that relies on freely available and abundant data (Tollefson, 2017). How both the community and the respective national space agencies respond to these opportunities (and risks) will go some way to

defining the direction of hydrological (and related) sciences over the next decade and beyond. Given our stakeholder position and vested interest in this, it would make sense to help shape the direction of these seemingly inevitable developments.

### 4.3     Continuity and Stability or Disruption and Opportunity

As has become clear, there are exciting future opportunities for hydrologic science that do not rely upon traditional space-borne approaches. The recent advent of low-cost UAVs, smartphones, and the global internet empowers the individual

researcher to collect their own measurements and hence to drive and direct their own scientific goals. For instance, scientists and engineers are no longer reliant on space agency airborne campaigns that can take years to organize, cannot respond to fast-paced dynamic events (such as floods, droughts, extreme events) and are always subject to the meteorological vagaries of the planned in advance experimental window (e.g., soil moisture campaigns that do not rain). But these approaches remain largely process-based and local in scale, so determining whether or how they can they be scaled-up to regional programs is

an important objective. Likewise, and perhaps more importantly, ensuring that these distributed and often uncoordinated efforts can be more closely tied to existing space-based measurements or local-to-global monitoring programs is an issue requiring community attention. To narrow the scope of this discussion somewhat, we revisit the changing role and nature of space based sensing: particularly the opportunities and challenges that will be presented from the rise of commercial sensing.

Hydrologists, like all scientists, need measurements, models, and money to make discoveries. From our review, it seems inevitable that at least for the immediate (and somewhat) foreseeable future, there will be positive and negative outcomes for the EO community with both technological changes and new players entering the space-based observation sphere. Although government agencies are unlikely to radically alter their EO programs, barring some unforeseen political event or paradigm





shift (a positive), the moneys that space agencies receive have remained historically flat, while costs continue to rise (a negative). So, while the positive enables a significant sized research community, the negative is that there will likely be fewer satellites and hence a lower variety of needed measurements available to advance our understanding of the Earth system. Space agencies will surely do their best to continue funding for individual research communities e.g. working groups

and airborne campaigns for each unique sector studying their particular component of the water cycle, and such approaches may well lead to scientific discoveries. But these will inevitably be at local scales and not at the global scale that satellites are designed to address. Moreover, while the traditional space agency approach of a careful and often prolonged mission planning and approval schedule (the counter example to fast and nimble) may lead to the eventual launch of a satellite measuring one aspect of the water cycle, there is no guarantee that other components will be simultaneously retrieved, and

hence the error envelope of models (and observations) will remain unconstrained. One of the outstanding challenges of hydrological remote sensing remains to monitor (and close) the water cycle (McCabe et al., 2008; Sheffield et al., 2009; Zhang et al., 2016a), but an integrated water cycle observation strategy remains very much in the conceptual phase, with no planned mission on the horizon.

Over the last few years, the commercial sector has demonstrated that space is now "open for business". A singularly positive outcome of this is that there now exists a range of near-daily to daily global VNIR band measurements that are available (albeit at a cost) from the commercial sector, providing ultra-high resolution detail. These commercial sensors provide data at a higher spatial and temporal resolution than comparable space agency systems (Dash and Ogutu, 2016), although the radiometric quality of the imagery may not always be as refined (Houborg and McCabe, 2016). As already noted, there is

generally no underlying scientific purpose or social good directly driving these efforts: commercial launches are ultimately driven by an economic incentive. As such, one negative resulting from this misalignment of purpose is that sensors that do not have an obvious income generating market are unlikely to be launched. For instance, active sensors have yet to make commercial inroads in the same way as optical sensors, and thus water cycle measurements that rely upon lidar or emitted radar pulses are not presently available (n.b., UrtheCast plan to equip their next generation satellite with an X- and L-band

active radar in early 2017, see Becket, 2015). But profit incentive is not the only difference separating these competing interests. Space agencies and the communities they serve often have an interest in data continuity: indeed, the Landsat mission has a legislated foundation to provide data "sufficiently consistent (in terms of acquisition geometry, coverage characteristics, and spectral characteristics) with previous Landsat data to allow comparisons for global and regional change detection and characterization" as part of the 1992 Land Remote Sensing Policy Act (US. Code Title 15, Chapter 82) (Irons

et al., 2012). In light of technological advances (e.g. constellations of CubeSats) and other space agency sensors such as Sentinel-2, it could be argued that continuity of a particular mission or sensor type is no longer necessary, so long as the observations lack discontinuities caused by large spatio-temporal gaps or calibration issues. The point here is that unlike the scientific community, the commercial sector has no demand or underlying rationale for ensuring continuity beyond satisfying the needs of their particular business model. Likewise, if there is an economic incentive to pursue it, they can





move quickly from one technology to the next without concern for the integrity of the long term data record: a position that may not be as easily adopted by space-agencies. Of course, a potential drawback of commercialisation lies in the quality and assessment of the delivered products. While many space agencies now allocate a proportion of the mission budget for cal/val related activities, this is not an aspect that would necessarily be considered by commercial ventures. The consequence of less

stringent quality controls is that any data from new commercial platforms may contain poorly-defined accuracies and sensitivities, hampering the process of time-series and multi-satellite data merging.

Given the somewhat meandering nature of research to applications, the commercial model may not seem to have immediate relevance to advancing scientific inquiry. However, there is much to be gained in leveraging and engaging with the influx of

activity in the current race to space, particularly given the range and variability of measurements that can provide new insights into process scale and response and with a density and fidelity that has never been seen before. One aspect that is not clear is whether the commercial sector will ultimately be in competition, or in cooperation, with government funded space agencies. Noting that both groups provide VNIR band imagery, it might seem that they are marketing the same product. Indeed, from an economics perspective, competition usually lowers costs. But given that space agency data are largely "free"

to the scientific community (n.b., this ignores the very real cost of tax-payer funded mission launches and data collection, processing and archiving), there would not seem to be any competitive advantage or level playing field. Clearly, the value proposition will be in resolution, timeliness, or in value adding i.e. increasing imagery information content through derived or customer specific products. How government space agencies might adapt to account for this commercial rise is unclear. There are threats, but also opportunities, particularly in the demonstration of new technologies and rapid delivery of

payloads to space. There are also obvious risks in a solely commercially driven framework: uncertainties in financing, profit making incentive, imagery costs, free-use policies and freedom to publish are all potential inhibitors to unhindered scientific inquiry. The future is certainly not clear. But these are issues that require immediate consideration given what seems to be an inevitable advance towards a greater commercialisation of Earth observation.

## 5    Concluding Remarks

We have entered a new era of Earth observation, where the threshold for what can be sensed from small satellite, airborne platforms or even on-ground monitoring is rapidly changing and evolving. The EO technologies discussed throughout this synthesis show great potential to revolutionize and reinvigorate our understanding of hydrology and present the community with exciting platforms from which to develop new insights into process form and function. The community has an opportunity to reshape hydrologic science across the spectrum, from fundamental research to applications based objectives.

Either in isolation or (ideally) in combination, researcher-led, commercial and government driven EO enterprises present new and innovative ways to envision the hydrological sciences and related disciplines. While there are certainly challenges in realising the potential of these emerging technologies, there are game-changing opportunities as well, from the novelty of





new sensing platforms such as CubeSats and UAVs, to the reshaping of the computational landscape through cloud-computing and data-analytic approaches. It is our hope that this forward-looking synthesis article will accelerate this (r)evolutionary process.

The alignment of circumstance and technology driving these advances have not happened in isolation, but reflect a convergence of innovation, breakthroughs in computational infrastructure and data storage, and opportunities for leveraging public and private assets collectively. Many of the EO advances discussed herein have arisen in just the past five years. What might the next 5-10 years have in stall? One possible scenario is contingent on the provision of global and low cost internet access (see discussion in Sect. 3.3 and efforts such as http://oneweb.world/). Given some notable failures of previous

attempts, the following remains rather speculative, but presents a plausible vision of the future. With an ever-increasing availability of low cost sensors, the connectivity provided by a global internet would facilitate truly autonomous remote monitoring of the Earth system. Whether permanent, disposable or even biodegradable, thousands of cheap devices could be deployed to measure soil moisture, precipitation, snow, stage or any other imaginable variable (see van de Giesen et al., 2014), recording and broadcasting directly to the internet or through scheduled collection via targeted UAVs or sentry-

systems (balloons, solar planes) in more remote regions. In such a connected world, integrating these diverse EO sources, from space-based to *in situ*, in order to optimise observing potential will be a key challenge. Technology is not the barrier to realising such a future, as much of what is needed exists already. But embracing these technologies will require a radical rethink, not just on how data is collected, but how it is used and managed in our modelling and interpretation efforts, where the focus on point-precision accuracy and error quantification can act as barriers to broader system understanding. What is

already evident is that humans have the capacity to traverse all corners of the globe and have the technology required to measure or infer most variables of interest. It is possible that *we* may be the remote sensing platforms of the future.





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



**Table 1. Hydrological variables and the current and planned satellite remote sensing missions that can be used to estimate them. We note that this list is not necessarily comprehensive and that there are possible trade-offs between resolution and accuracy that are not explicitly accounted for in this list.**

| Hydrological Variable | Missions / Instruments | Standard Spatial Resolution (km) | Standard Temporal Resolution (days) | Launch Year | Dedicated Measurement |
|---|---|---|---|---|---|
| Rainfall | GPM | 5 | 0.125 | 2014 | Y |
| Snowfall | GPM | 5 | 0.125 | 2014 | N |
| Evaporation | Terra/MODIS Aqua/MODIS Suomi/VIIRS | 0.5 | 1 | 1999 2002 2013 | N |
| | Landsat 8 Landsat 9 | 0.03 | 32 | 2013 2023 | N |
| Runoff | SWOT | 0.1 | 11 | 2021 | Y |
| Snow Cover | Terra/MODIS Aqua/MODIS Suomi/VIIRS | 0.5 | 1 | 1999 2002 2013 | Y |
| Snow Density, Depth, or Water Equivalent | GCOM-W/AMSR2 | 30 | 1 | 2012 | N |
| Surface Soil Moisture | SMOS | 36 | 3 | 2009 | Y |
| | SMAP (radiometer) | 36 | 3 | 2015 | Y |
| | ASCAT | 25 | 1 | 2006 | N |
| | GCOM-W/AMSR2 | 50 | 1 | 2012 | N |
| | Sentinel-1A Sentinel-1B | 0.1 - 0.005 | 12 | 2014 2016 | N N |
| Deep Soil Moisture | Biomass | 0.2 | 18 days/yr | 2021 | N |
| Surface Water Elevation | Jason-3 | 10 | 10 | 2016 | N |
| | SARAL | 10 | 35 | 2013 | N |
| | SWOT | 0.1 | 11 | 2021 | Y |
| | ICESat-2 | 1.5 | 90 | 2018 | N |
| *Depth to Groundwater* | - | - | - | - | - |
| *Total Groundwater Storage* | - | - | - | - | - |
| Terrestrial Water Storage Change | GRACE | 220 | 30 | 2002 | Y |
| | GRACE-FO | 180 | 30 | 2017 | Y |
| *Water Consumption* | - | - | - | - | - |
| *Water Quality* | - | - | - | - | - |
| Vegetation/Land Cover/Irrigated Area | Terra/MODIS Aqua/MODIS Suomi/VIIRS | 0.5 | 1 | 1999 2002 2013 | Y |
| | Landsat 8 Landsat 9 | 0.03 | 16 | 2013 2023 | Y |
| | Sentinel-2A | 0.02 | 10 | 2015 | Y |
| | Sentinel-2B | 0.02 | 10 | 2017 | Y |
| | Sentinel-3A | 0.3 | 2 | 2016 | Y |
| | Proba-V | 0.35 | 2 | 2013 | Y |
| Vegetation Stress | ISS/ECOSTRESS | 0.07 | 4 | 2018 | Y |
| Photosynthesis | FLEX | 0.3 | 0.5 | 2022 | Y |
| Water Vapour | Aqua/AIRS | 13.5 | 1 | 2002 | N |
| *Integrated Water Budget* | - | - | - | - | - |





**Figure Captions**

**Figure 1. The state of play in space today. Estimates are based on the Union of Concerned Scientists satellite database, updated from 30/6/2016 (see http://www.ucsusa.org/nuclear-weapons/space-weapons/satellite-database). In terms of the sectors operating**
**Earth Observing systems (right panel), another 5% include shared systems between those listed.**

**Figure 2. An Earth observing "System of Systems" for revolutionizing our understanding of the hydrological cycle. This multi-scale, multi-resolution observation strategy is not a concept: the technology exists and is largely in place now. Supporting traditional space based satellites, there are now a range of orbital options from commercial CubeSats to demonstration sensors on-**
**board the International Space Station. Beyond orbiting EO systems, technological advances in hardware design and communications are opening the skies to stratospheric balloons and solar planes, as well as an explosion of UAV-type platforms for enhanced sensing. At the ground level, the ubiquity of mobile devices are expanding traditional in-situ network capacity, while proximal sensing and signals of opportunity are opening up novel measurement strategies.**

**Figure 3. Employing a UAV to retrieve high-resolution multispectral information on the land surface for hydrology and related applications over an Australian rangeland site located near Fowler's Gap in New South Wales. Retrieved products include: a) a false-colour infrared image; b) a reconstructed digital surface model using visible imagery and structure-from-motion techniques; and c) an optimized soil adjusted vegetation index (OSAVI) derived from the 4-band multispectral image. Images were captured using a MicaSense/Parrot Sequoia sensor on-board a 3DR Solo quadcopter. The UAV was flying at a height of 40 m, providing a**
**ground sampling distance of approximately 3 cm. Imagery provided by the University of Tasmania's TerraLuma Research Group.**

**Figure 4. Multi-scale capabilities of state of the art sensing optical satellites. Image illustrates the expanding resolution options available from both commercial and government satellites. A) Planet CubeSat at 3 m ground sampling distance over the Tawdeehiya Farm in Al Kharj, Saudi Arabia. Center pivot irrigated fields dot the landscape, with dimensions approaching 800 m.**
**The inset in A) is zoomed to show the resolution advantages offered by the next generation of sensing solutions over B) Landsat-8 at 30 m, with C) Sentinel-2A at 10 m and D) Planet imagery at 3 m providing enhanced details. All images are false colour representations of NIR, Red and Blue in RGB bands. Sentinel-2A and Landsat-8 images were acquired on December 4th, 2016, while the Planet data were captured on December 5th, 2016.**

**Figure 5. Worldwide global system for mobile communication (GSM) coverage for the year 2013. The GSM network does not include the growth of related 3G or 4G networks. The image is derived from Figure 2 in Overeem et al. (2016).**

**Multimedia 1. On-board the International Space Station, the Urthecast IRIS high-resolution camera (HRC) capture colour video at 3 frames per second for a duration of 60 seconds. Here we see an example of the HD Video over the Burj Khalifi in Dubai. The**
**tracking of vehicles on roads is analogous to monitoring flow in rivers or the speed of moving clouds, while the capacity to extract 3D structure of the underlying terrain provides opportunities in dynamic monitoring of surfaces.**





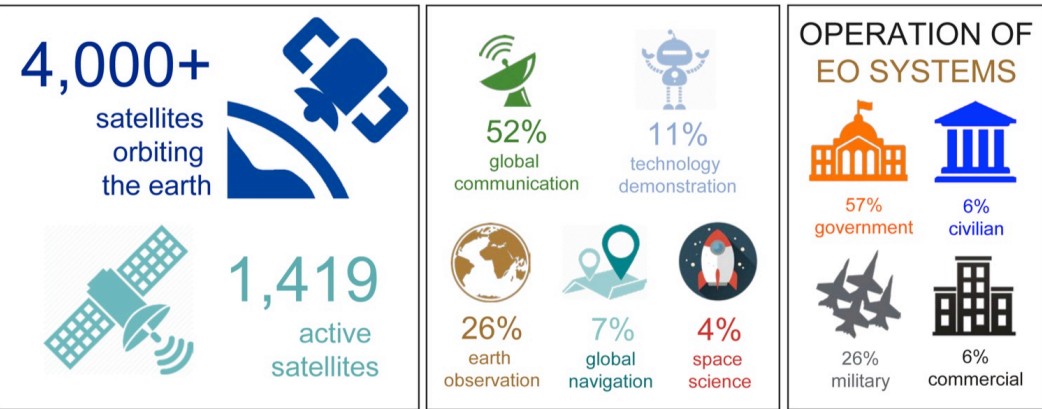

**Figure 1. The state of play in space today. Estimates are based on the Union of Concerned Scientists satellite database, updated from 30/6/2016 (see http://www.ucsusa.org/nuclear-weapons/space-weapons/satellite-database). In terms of the sectors operating Earth Observing systems (right panel), another 5% include shared systems between those listed.**





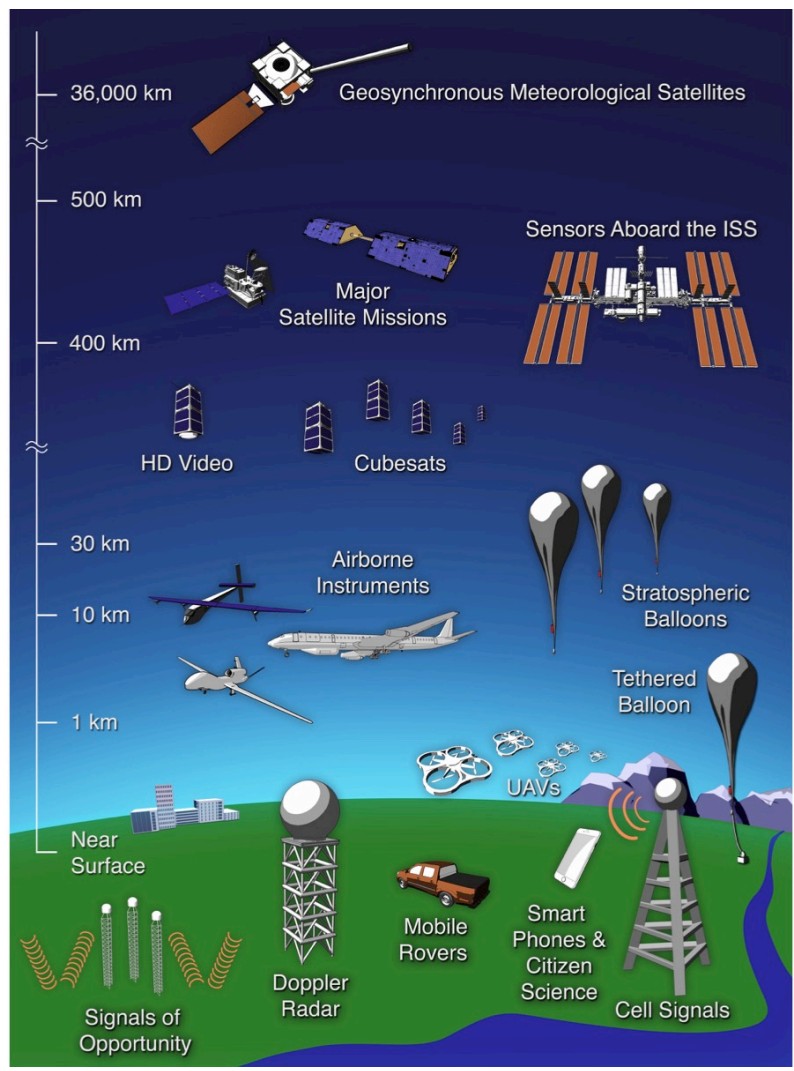

**Figure 2. An Earth observing "System of Systems" for revolutionizing our understanding of the hydrological cycle. This multi-scale, multi-resolution observation strategy is not a concept: the technology exists and is largely in place now. Supporting traditional space based satellites, there are now a range of orbital options from commercial CubeSats to demonstration sensors on-board the International Space Station. Beyond orbiting EO systems, technological advances in hardware design and communications are opening the skies to stratospheric balloons and solar planes, as well as an explosion of UAV-type platforms for enhanced sensing. At the ground level, the ubiquity of mobile devices are expanding traditional in-situ network capacity, while proximal sensing and signals of opportunity are opening up novel measurement strategies.**



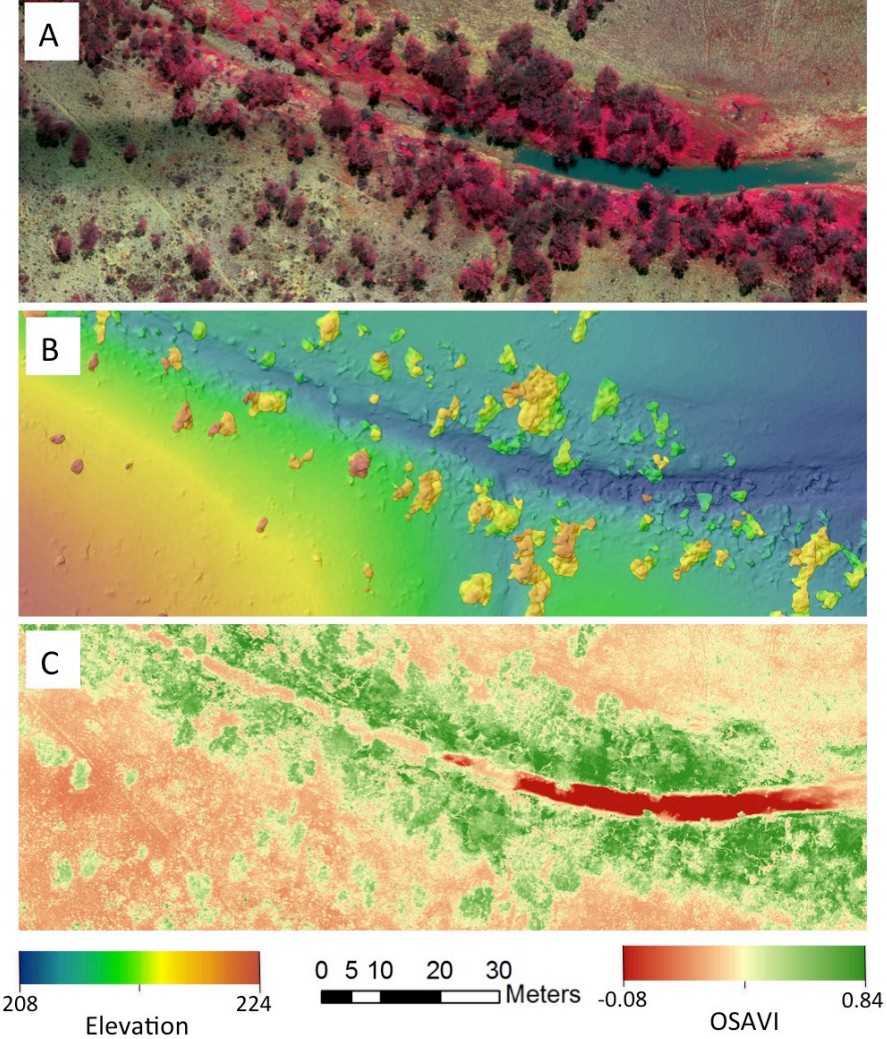

**Figure 3. Employing a UAV to retrieve high-resolution multispectral information on the land surface for hydrology and related applications over an Australian rangeland site located near Fowler's Gap in New South Wales. Retrieved products include: a) a false-colour infrared image; b) a reconstructed digital surface model using visible imagery and structure-from-motion techniques; and c) an optimized soil adjusted vegetation index (OSAVI) derived from the 4-band multispectral image. Images were captured using a MicaSense/Parrot Sequoia sensor on-board a 3DR Solo quadcopter. The UAV was flying at a height of 40 m, providing a ground sampling distance of approximately 3 cm. Imagery provided by the University of Tasmania's TerraLuma Research Group.**





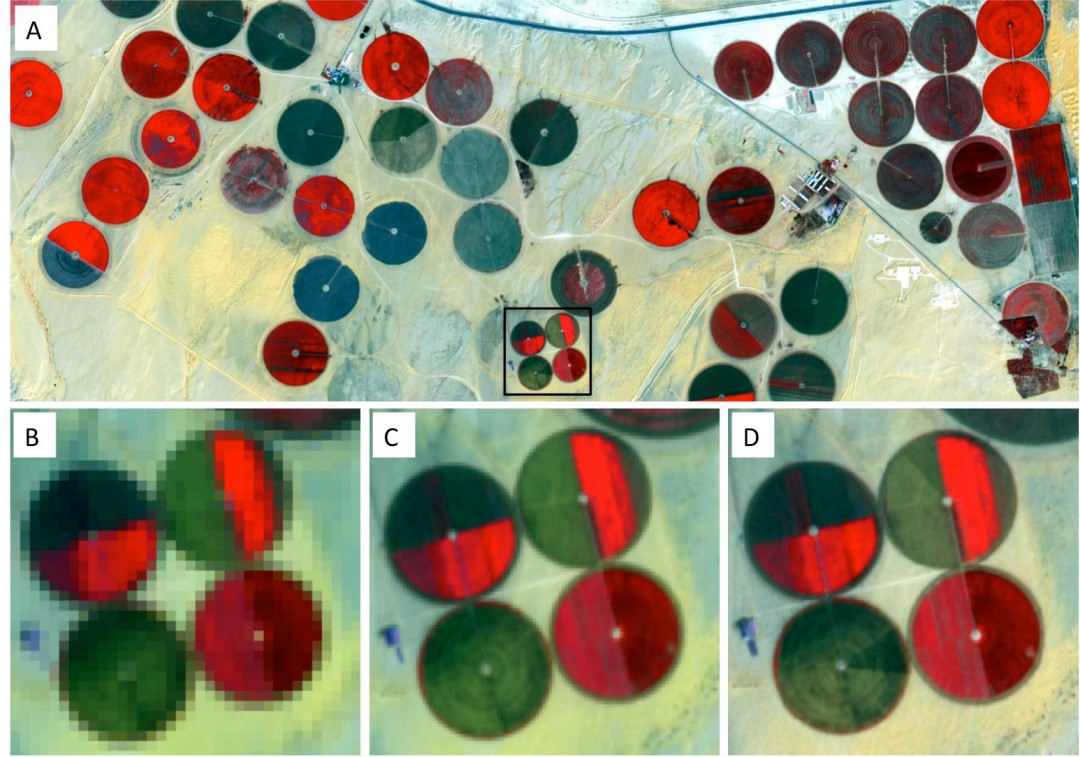

**Figure 4. Multi-scale capabilities of state of the art sensing optical satellites. Image illustrates the expanding resolution options available from both commercial and government satellites. A) Planet CubeSat at 3 m ground sampling distance over the Tawdeehiya Farm in Al Kharj, Saudi Arabia. Center pivot irrigated fields dot the landscape, with dimensions approaching 800 m.**
5 **The inset in A) is zoomed to show the resolution advantages offered by the next generation of sensing solutions over B) Landsat-8 at 30 m, with C) Sentinel-2A at 10 m and D) Planet imagery at 3 m providing enhanced details. All images are false colour representations of NIR, Red and Blue in RGB bands. Sentinel-2A and Landsat-8 images were acquired on December 4th, 2016, while the Planet data were captured on December 5th, 2016.**




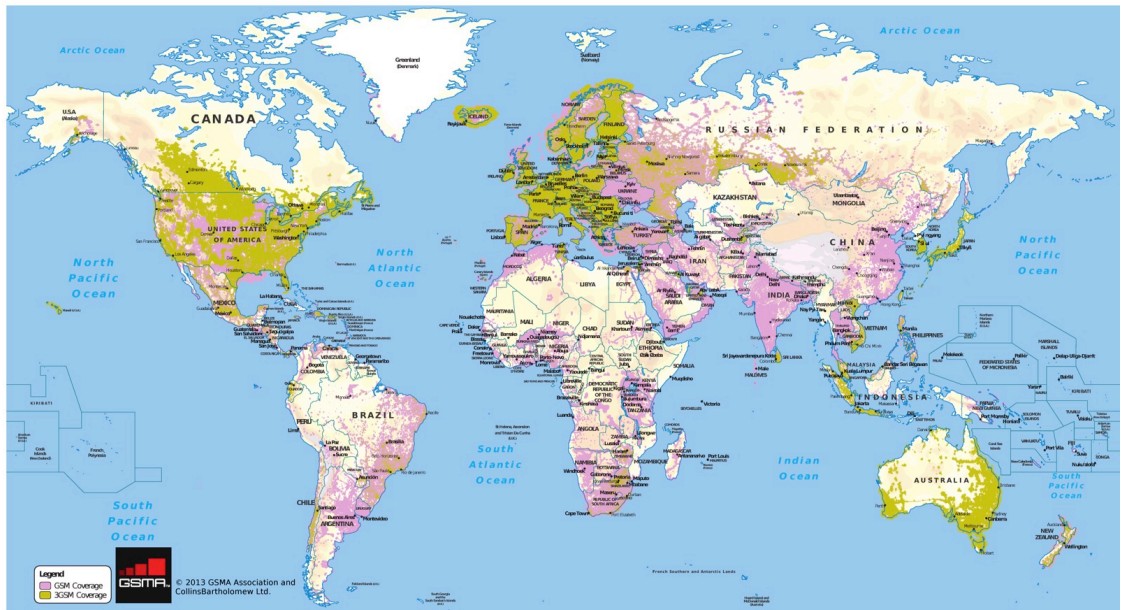

**Figure 5. Worldwide global system for mobile communication (GSM) coverage for the year 2013. The GSM network does not include the growth of related 3G or 4G networks. The image is derived from Figure 2 in Overeem et al. (2016).**





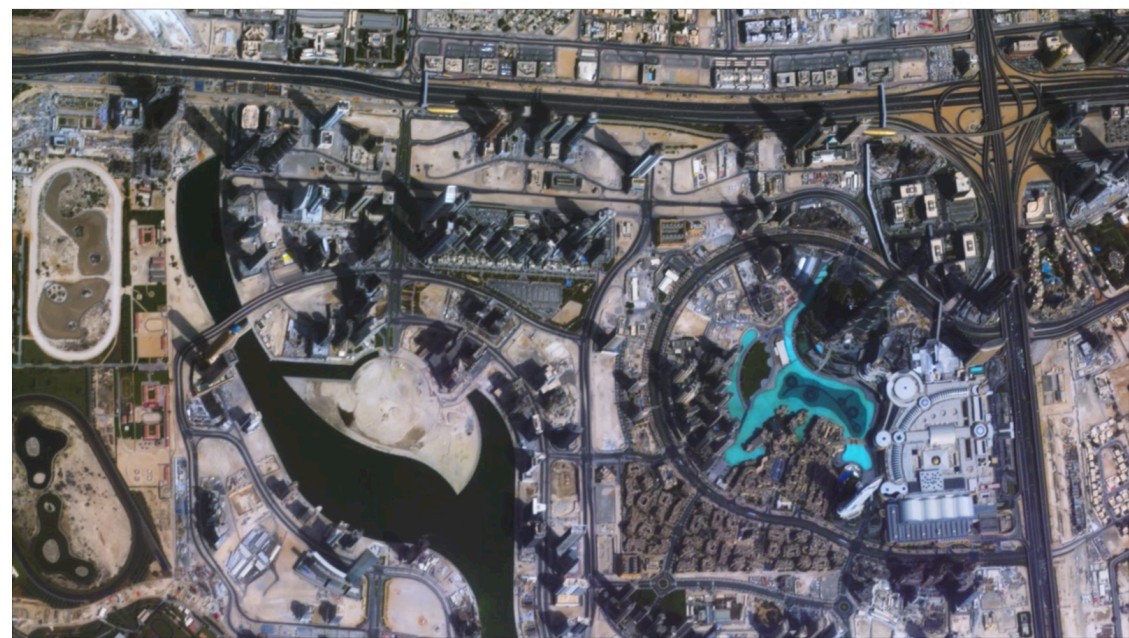

**Multimedia 1. On-board the International Space Station, the Urthecast IRIS high-resolution camera (HRC) capture colour video at 3 frames per second for a duration of 60 seconds. Here we see an example of the HD Video over the Burj Khalifi in Dubai. The tracking of vehicles on roads is analogous to monitoring flow in rivers or the speed of moving clouds, while the capacity to extract 3D structure of the underlying terrain provides opportunities in dynamic monitoring of surfaces.**