# Peer review of "The Future of Earth Observation in Hydrology"

_Hydrology and Earth System Sciences, 2017_

## Short Comment (SC1) · 18 Feb 2017

McCabe et al provide a well-written account of a multitude of options for remote sensing of hydrology. The paper, however, is framed squarely in an us-versus-them: commercial versus governmental agencies. This framing is oft used by the commercial sector at this stage, as government agencies present the most direct competition to business. So, we hear a lot about how commercial companies do more for cheaper and faster, etc.; but, this comparison can be very misleading. They are not doing the exact same things, so the comparison may be flawed. E.g., budget comparing GRACE to multiple VIS's. The combination of spectral range, accuracy, durability, quality, spatial coverage, data continuity, and transparency is limiting with current commercial capabilities. The authors do discuss this, but only after glowing about the commercial sector. At one point, McCabe et al state, "The commercial model may not seem to have immediate

relevance to advancing scientific inquiry." A comment on the article title—it is not well aligned with the paper content, which focuses on commercial technology (though reviewing very well and thoroughly governmental capabilities). A more accurate title would be, "The Future of Commercial Earth Observation in Hydrology,"; or, possibly: "The Future of Commercial Earth Observation in Hydrology: the commercial model may not seem to have immediate relevance to advancing scientific inquiry"... ;)

Along these lines, what is desperately needed for the authors to make their case more compelling is to start with the key science and applications needs, the necessary observational requirements, the associated technological capabilities, and then map those onto commercial capabilities (and maybe potential business/market case). For instance, if we require high spatiotemporal resolution permafrost coverage of the pan-Arctic, then are drones going to do it? Can cubesats handle thermal radiometers or LiDAR? It's not that drones aren't useful–it's that they are useful for specific applications. The text should be structured more along the lines of that traceability matrix.

I do agree with the authors that the commercial sector has enormous potential to add to and advance the state of hydrologic remote sensing. But, the case needs to be made a bit more measured and analytically, staying away from the "commercial sector can do everything governmental agencies can do, just cheaper, faster, and better" marketing lines.
* * *

---

## Referee Comment (RC1) · Anonymous Referee #1 · 3 Mar 2017

This paper is an admirably comprehensive overview of how changes in sensing platforms, data providers of spaceborne applications, enable new data sources, and thus, presumably, new breakthroughs in hydrology. No single paper can be expected to comprehensively cover the entire possible 'future of hydrology' and thus this provides a somewhat biased view (about which a caveat in the text would be useful). Nevertheless, the authors have done a remarkably thorough job in terms of capturing the various exciting opportunities available. I particularly appreciate the high number of references across such a diverse paper. Nevertheless, some increased focus and streamlining would significantly improve the paper and is necessary. Many of the points made in the current paper are likely to be lost on the reader without it.

Specific Comments: 1) Section 2 isn't clearly tied to the rest of the paper. Section 2.1 is a haphazard mix of challenges that can be addressed by using a wider diversity

of platforms, commercial or otherwise, highlighted in Sections 3 and 4 (e.g. the risk of launches and focusing on single large instruments), and ones that are unrelated (e.g. greater multi-mission integration could be achieved also with space agencies and traditional large LEO or geostationary satellites), and ones that may even be made worse if traditional EO platforms break down (e.g. the perception that remote sensing data are magic and ignorance about reliance on retrieval algorithms may become worse with people using more commercial solutions that, for reasons of competition with other companies, are far more likely to keep their technological and retrieval details secret). Additionally, the challenges and opportunities mentioned for each variable in Section 2.2 are somewhat haphazard. It would be useful to specifically link each of the variables to opportunities mentioned in the rest of the paper.

2) Section 4 comes across too much as an advertisement for EO start-ups in Silicon Valley and a list of problems with space agency-built sensors (as also mentioned in Josh Fisher's non-reviewer comment). The challenges associated with commercial data are mentioned either as an aside that belies the serious nature of the issue (unknown and most likely lacking calibration relative to scientific standards in most cases) , or not at all (data continuity, data costs, the fact that Cubesat orbits aren't always perfectly predictable a priori because of the way in which they are launched). The lower budgets of these startups are presented as being due to the nimbleness of commercial industry, but don't reflect the fact that launching lots of small satellites is inherently cheaper, or that SpaceX has relied on massive amounts of capital investments and has a high failure rate. This is a dis-service not only to government agencies as a source of EO data, but also to the points raised in Section II (which are lost in the process) and to the fact that, in reality, and the fact that future models and analyses efforts are likely to be best aided by a combination of both single-sensor satellites that provide reliable data at perhaps coarser spatial resolution and high resolution data from Cubesats. This section would be significantly aided by a summary table of pros and cons for a) government vs. commercial observing systems. The enormous potential of commercial systems (rightfully highlighted) would go to waste if the community cannot also

think and overcome their related challenges. If space is an issue, such a table would therefore be more useful than e.g. table I, which summarizes established knowledge.

Relatedly, the authors may also want to consider renaming the paper "Opportunities and challenges for the future of remote sensing" to further highlight that a world where hydrologists predominantly use data from the start-ups and new platforms identified in this paper would be great but is not guaranteed if the challenges mentioned throughout are overcome.

3) Section 3.2 only has one subheading, and it is unclear why UAVs deserve their own subheading but balloons and solar planes do not.

4) It would be useful to have a short summary section somewhere that explicitly summarizes challenges to the community and how it operates. E.g. the need to move towards cloud more, publish processed datasets to the same cloud platform rather than keeping them behind user accounts, think about how to engage with commercial agents, building data ingestion systems that are based on more haphazard temporal and spatial coverage, etc. Such issues are now mentioned as asides throughout and may seem less important, although they are arguably among the most potentially actionable results of the paper.

5) Several sections have noticeably lower added value than others, and may be best deleted to keep the paper length manageable and make the rest of the paper clearer. For example, the historical introduction about ground measurements in Section 3.6 (p. 25) only barely links up to the rest of the section (except maybe through the "Internet of Things" reference, but even that is only useful to someone who has already spent time thinking about the IoT and will grasp the connection immediately). Similarly, discussion of the declining trend in ground observations in Section 2.1 breaks the flow and is somewhat out of place. The section on potential new airborne measurements for commercial aircraft (p. 27) is far too vague – what variables could possibly be easily added? Airborne instrumentation for radar, fluorometry or even optical measurements

requires extensive retrofitting of the aircraft and e.g. instrument protection in the form of a radome. By contrast eddy covariance measurements are difficult to calibrate.

Minor Specific Comments:

* Section 3 title: It would be useful to include the phrase "data sources" or "platforms" somewhere in this section title, as that is the focus of many of the sub-headings.

* Page 4, Line 9: The A-Train efforts and large number of sensors aboard Aqua have been useful so some care with phrasing is needed here.

* Page 7, Point 4: The first sentence here provides far too limiting a view of the possible uses of hydrological data. It does not include, for example, the use of hydrologic earth observation data (without further assimilation) in carbon-cycle or biogeochemical studies, epidemiological ones, or even ones on social science implications of variations in water availability (e.g. Muller et al, PNAS, 2016 "Impact of the Syrian refugee crisis on land use and transboundary freshwater resources"). It would be more correct to delete the first sentence and rephrase that the ability to have long-term trends is useful for many studies instead of being so definitive.

* Page 7, Line 19: Su et al, Geophysical Research Letters, 2016 ("Homogeneity of a global multi-satellite soil moisture climate data record") may be a helpful citation in the discussion of multi-sensor merging

* Page 12, Line 3: Note that e.g. Konings and Gentine, Global Change Biology, 2017 ("Global variations in ecosystem-scale isohydricity") is a noteable exception here.

* Page 15, line 14: NASA/DLR rather than NASA/German

* Page 36, Line 13: The Chinese WCOM mission may be worth mentioning as an exception here, though it is still in the early phases

* You may want to mention that Planet was recently sold

* The 220 MB multimedia supplement may not be worth its large file size. Although it is

difficult to note the cars moving unless looking for it. If the authors really want to include it, it may be worth cropping to a smaller area that focuses more on the highways.

* A few informal abbreviations have slipped in throughout: tech rather than technolog, 'till, jet Propulson lab rather than laboratory, etc..

---

## Short Comment (SC2) · 21 Mar 2017

C. Kummerow

Kummerow@atmos.colostate.edu

Overall, I find the paper suitable for publication in that I though it did a good job at reviewing the current space-based observing systems, and the emerging capabilities and technologies. I found the last section a bit unbalanced as I explain below.

The author lists existing and planned observations together with a comprehensive look at emerging technologies to argue that the future of hydrology is not in individual datasets but instead in the integration of large clouds of emerging data sources. The author makes a good case that the emerging data sources indeed hold promise and that storage and computing may be available as well.

Where the paper falls short, in my view, is in assuming that if the pieces exist, the implementation is easy. Space agencies currently devote a significant amount of their

funding to integrate satellite measurements into a coherent global framework. As more and more data sources shift the focus away from specific space agencies, it would be hard to imagine that they would be able to fund such large data storage and management objectives unless they are tied directly to their mission. The real question becomes – who funds such a large and integrating hydrological sciences data structure objective? The answer is that nobody is in a position to step up to that task at the moment, and without it, the holistic approach is not in easy reach.

I don't think the authors need to answer this question. On the other hand, it is an immense challenge that cannot simply be ignored. The challenge before us is not simply in making measurements, but also rethinking how we work as a community and how we fund this community. That needs to be better conveyed in this paper.

Some places where corollaries of this show up: p. 5, line 1: the bottom of page 4 argues that it would be great if there was a continuous, holistic water budget strategy. The author then suggest that aside from a paradigm shift in how we undertake much of our research, the big hurdle would be the need to have much more and much cheaper observations. Lots of cheap observations are a problem for space agencies who's mission it is to advance space-borne technology.

p. 7, line 10: The notion that we need a more appropriate error analysis before we can merge multiple sensors into a coherent framework is often repeated and yet to be implemented. Since this is a large part of what the paper is advocating for the future, the authors should emphasize the critical need to advance this field rather than simply stating it as a necessity.

p. 29, line 22: Google's Earth Engine was developed by Google and follows Google's rules. Without a coordinating body in global hydrology, it becomes difficult to reproduce something like this.

p. 34, line 27: The fallacy here is that it assumes that if SpaceX develops a very cheap launch vehicle, then NASA would launch its satellites on it. That is probably not the

case. NASA would continue to develop new technologies that in its early stages, might be just as expensive as before.

p. 30, line 22: Storage can get very large. While commercial vendors often discard old data which generates little revenue, this has never been the case in Earth Sciences. It again goes back to the question of funding data storage that belongs to the community rather than an "agency".

Some minor text issues:

Abstract, line 10: The real time, high resolution video capabilities are surely new, but equally important for cloud development are the newly available JMA and NOAA 1min. geostationary data with 1 km resolution. CMA even has 60m resolution geostationary data with 30 sec. resolution. Given the size and speed with which clouds naturally evolve, it is not clear to me that 10 m data with 30 frames per second would revolutionize our understanding of clouds. I can better see the utility in tracking small scale, fast flows. In any case some careful rewording here would help.

p. 6, line 5: Eliminate "radiative" – just visible and NIR frequencies is enough.

p. 6, line 6: It is now GOES-16 and authors should probably mention Himawari-8 as well. They do the same thing.

p. 9, Line 16: Doppler radars are good for judging speeds and rotation of convective elements. They don't do anything for rainfall estimation. I suggest that the author drop "Doppler" or replace it with "polarimetric".

p. 14, Line 5: Section 2 discussed variables independently. The author mentions that integration and a more holistic approach is necessary – but emergent capabilities and technologies are not the answer. They merely add observations. The text is not very clear here.

p. 14, line 11: The text starts out boldly predicting hydrology 50 years from now. The paper, however, focuses on things that are real today and probably represents the next

10-15 years. Perhaps that would be a better into?

p. 14: Line 33: Perhaps Japan should be mentioned as well.

p. 18, line 25: The balloon section needs a few more caveats. Since high altitude balloons have been around for a long time, and the authors fail to make a case that commercial balloons for telecom would somehow change their utility or cost, some argument for why we might see a resurgence of balloon-based observing systems should be put forth.

p. 18, line 32: "constructed" instead of "constructing".

p. 21, line 19: perhaps the authors need to still remind readers that while passive systems in the VIS/IR are quite feasible, the power needed by active sensors still limits these systems today.

p. 23, line 5: I think the authors could come up with a better example here. Rain gauge records are often protected for various geopolitical reasons. It is not clear to me that the rain gauge records from the Congo have not been released simply because nobody in the office has a smartphone.

p. 24, line 12: The authors might want to include CoCoRAHs https://www.cocorahs.org/ Which, on a given day, receives roughly 7000 daily rain gauge reports) from citizen scientists.

p. 25, line 20: Perhaps it is worth noting that the "entire country" was the Netherlands and that this would not work for countries like the US where many rural areas do not have nearly enough cell phone towers to apply this technique.

p. 27, line 3: Commercial aircraft started making turbulence measurements as well. See Sharman et al., Description and Derived Climatologies of Automated In Situ Eddy-Dissipation-Rate, 2014. Appl. Met. and Clim., 53, 1416-1432

p. 27, line 25. The same issue pointed out in the abstract.

[Figure]

p. 33, line 4: The risk averse nature of space agencies is true. The rationale is the authors' interpretation. I would argue that they are risk averse because it takes so long to reach scientific consensus. The PI led missions have very short time frames – similar to industry. Perhaps pointing out that facility type missions are risk averse while PI missions can be much more nimble is a better way of expressing this.

p. 35, line 20: At least NASA is shifting into smaller missions via Earth Ventures. This should be acknowledged although it is equally true that NASA's mission would probably not support a much greater emphasis in small Earth Venture missions.

---

## Referee Comment (RC2) · G. D. Salvucci (Referee) · 13 Apr 2017

I recommend publication possibly subject to very minor revisions.

First off, the paper is excellent and extremely valuable to the community, and, frankly, though provoking.

I honestly had no idea about more than half of the issues raised in the paper. I can imagine that, for younger scientists/engineers, both hydrologic practitioners and researchers, the paper will be especially motivating. With faculty positions and government research tight, the paper gives some hope for a larger industry including the private sector where many of our current and recent Ph.D. students could find employment, and most importantly, contribute significantly.

The paper is long, but that's ok. It is very well written.

I have only a few short comments that could be considered by the authors before publication:

1) In section 2.1, fundamental challenge #3, where stove-piping is criticized, i wonder if the EOS project isn't a counter-example (at least partly).

2) The point made in section 2.1, fundamental challenge #4, that due to merging "observed inter-annual fluctuations may reflect discontinuities in the constellation of satellites" is an important one, and thus could use a few citations, as this has been discussed a fair amount, and many merged products exist to attempt to overcome it.

3) in the future agency missinos discussion (3.1), the statement "Deep soil moisture could also be on the list, although soil moisture algorithms that make use of wavelengths longer than L-band are less than mature." is made without evidence. I honestly do not know the status of that research, but I support its potential importance, and am not comfortable with it being dismissed without backing.

4) Two sources of data that I was suprised were not discussed at all (or barely), are :

1) Ameriflux/Fluxnet (I beleive there was one citation to a paper that used fluxnet). I belive fluxnet has had a huge impact on hydrologic science, and hope to see it continue and grow in scope. I think it could be argued, as well, that for the cost of a space mission (billion dollars ?), one could put together a pretty amazing network of eddy covariance stations (perhaps 5,000 stations running for 5-10 years ?)

2) AMDAR/ACARS observations (e.g. Drue et al, 2008, QUARTERLY JOURNAL OF THE ROYAL METEOROLOGICAL SOCIETY Q. J. R. Meteorol. Soc. 134: 229–239 (2008)) which provide extremely high resolution profiles of temperature, wind and humidity that well sample the atmospheric boundary layer during takeoff and landing many hundreds of times per day all over the planet.

Signed: Guido Salvucci

---

## Author Comment (AC3) · 19 Apr 2017

We thank Prof Kummerow for his thoughtful interactive comment. A number of important points are raised in this review, particularly regarding the issue of integrating the diverse data sets and sources that are emerging into a "coherent global framework". This is most definitely a community challenge, but as the reviewer implies, it is beyond the scope of this paper to address. However, it is an important point that requires some comment in a revised version. A number of other excellent points are raised in the review, and we respond to these individually below.

p. 7, line 10: The notion that we need a more appropriate error analysis before we can merge multiple sensors into a coherent framework is often repeated and yet to be implemented. Since this is a large part of what the paper is advocating for the future,

[Figure]

the authors should emphasize the critical need to advance this field rather than simply stating it as a necessity.

[Response] The purpose of Section 2.1 is to highlight some of the challenges and knowledge gaps in remote sensing that require community attention. The particular challenge of multi-sensor integration and error characterization is a key example of these challenges. In revising the paper, we can certainly draw stronger attention to this issue, as we agree that it is a critical element of any comprehensive observation strategy.

p. 29, line 22: Google's Earth Engine was developed by Google and follows Google's rules. Without a coordinating body in global hydrology, it becomes difficult to reproduce something like this.

[Response] Agreed. But it is also difficult to imagine the scientific community organizing themselves and then funding an initiative to replicate such a framework. Whether it is necessary to do so, given that these systems are relatively easy to access and (currently) not especially restrictive, is also a consideration. There are examples of private-public partnerships that seek to reproduce some of the same functions as Google Earth Engine and Amazon Web Server (see discussion on EODC p.30, line 15) which may serve as a model should these commercial facilities prove unsuitable for scientific application.

p. 34, line 27: The fallacy here is that it assumes that if SpaceX develops a very cheap launch vehicle, then NASA would launch its satellites on it. That is probably not the case. NASA would continue to develop new technologies that in its early stages, might be just as expensive as before.

[Response] Hopefully, NASA will continue to develop new technologies, as the commercial sector are unlikely to invest the significant resources required to do this: a point mentioned in our paper. However, in terms of delivering satellites into space and re-provisioning ISS, NASA is already utilizing commercial sector expertise, and this model

seems likely to expand.

p. 30, line 22: Storage can get very large. While commercial vendors often discard old data which generates little revenue, this has never been the case in Earth Sciences. It again goes back to the question of funding data storage that belongs to the community rather than an "agency".

[Response] Yes, this is an important point, and there are examples within our own scientific communities where this has been done poorly. Further highlighting the importance of data archiving and stewardship can be addressed in a revised version of the paper. With the rise of AI and machine learning approaches, the importance of a long record of "training data" may be an incentive to effectively archive historical observations.

Abstract, line 10: The real time, high resolution video capabilities are surely new, but equally important for cloud development are the newly available JMA and NOAA 1min. geostationary data with 1 km resolution. CMA even has 60m resolution geostationary data with 30 sec. resolution. Given the size and speed with which clouds naturally evolve, it is not clear to me that 10 m data with 30 frames per second would revolutionize our understanding of clouds. I can better see the utility in tracking small scale, fast flows. In any case some careful rewording here would help.

[Response] Noted and we can revise the text (n.b. the HD video is on the order of 1m, not 10m). Perhaps "understanding" can be changed to "monitoring".

p. 6, line 5: Eliminate "radiative" – just visible and NIR frequencies is enough.

[Response] Noted

p. 6, line 6: It is now GOES-16 and authors should probably mention Himawari-8 as well. They do the same thing.

[Response] Noted

p. 9, Line 16: Doppler radars are good for judging speeds and rotation of convective elements. They don't do anything for rainfall estimation. I suggest that the author drop "Doppler" or replace it with "polarimetric".

[Response] Noted

p. 14, Line 5: Section 2 discussed variables independently. The author mentions that integration and a more holistic approach is necessary – but emergent capabilities and technologies are not the answer. They merely add observations. The text is not very clear here.

[Response] We agree that more observations will not directly resolve the issues raised in Section 2. The purpose of Section 3 (Emergent Capabilities and Technologies) is not to provide answers to these issues. A better transition paragraph to this section will be attempted in a revised version.

p. 14, line 11: The text starts out boldly predicting hydrology 50 years from now. The paper, however, focuses on things that are real today and probably represents the next 10-15 years. Perhaps that would be a better into?

[Response] The introductory line is definitely not intended to impart any bold or long term predictions of hydrology. Perhaps the comparison to the developments and improvements in our transportation system is not the clearest analogy. We will review this. Our focus is certainly more on the near-term (5-10 year horizon), highlighting technologies that for the most part have emerged within the last few years, but which we believe have the potential to play a much larger role in EO strategies and in advancing the EO landscape.

p. 14: Line 33: Perhaps Japan should be mentioned as well.

[Response] Noted

p. 18, line 25: The balloon section needs a few more caveats. Since high altitude balloons have been around for a long time, and the authors fail to make a case that

commercial balloons for telecom would somehow change their utility or cost, some argument for why we might see a resurgence of balloon-based observing systems should be put forth.

[Response] The argument here is that commercial enterprises (i.e. Google and Facebook) have active programs exploring the use of manoeuvrable high-altitude balloons and/or autonomous planes. This is driven by commercial motives for the most part. Whether our community can leverage this for scientific purposes remains unclear. We can more clearly articulate this in a revision if necessary, as well as some of the caveats.

p. 18, line 32: "constructed" instead of "constructing".

[Response] Noted

p. 21, line 19: perhaps the authors need to still remind readers that while passive systems in the VIS/IR are quite feasible, the power needed by active sensors still limits these systems today.

[Response] Noted

p. 23, line 5: I think the authors could come up with a better example here. Rain gauge records are often protected for various geopolitical reasons. It is not clear to me that the rain gauge records from the Congo have not been released simply because nobody in the office has a smartphone.

[Response] Noted. The political and bureaucratic issues are also worth highlighting.

p. 24, line 12: The authors might want to include CoCoRAHs https://www.cocorahs.org/ Which, on a given day, receives roughly 7000 daily rain gauge reports) from citizen scientists.

[Response] Thank you and noted.

p. 25, line 20: Perhaps it is worth noting that the "entire country" was the Netherlands

and that this would not work for countries like the US where many rural areas do not have nearly enough cell phone towers to apply this technique.

[Response] Noted.

p. 27, line 3: Commercial aircraft started making turbulence measurements as well. See Sharman et al., Description and Derived Climatologies of Automated In Situ Eddy Dissipation-Rate, 2014. Appl. Met. and Clim., 53, 1416-1432

[Response] Noted and we will include this reference.

p. 27, line 25. The same issue pointed out in the abstract.

[Response] Noted and we will highlight some additional aspects where HD Video may be better suited.

p. 33, line 4: The risk averse nature of space agencies is true. The rationale is the authors' interpretation. I would argue that they are risk averse because it takes so long to reach a scientific consensus. The PI led missions have very short time frames – similar to industry. Perhaps pointing out that facility type missions are risk averse while PI missions can be much more nimble is a better way of expressing this.

[Response] Noted.

p. 35, line 20: At least NASA is shifting into smaller missions via Earth Ventures. This should be acknowledged although it is equally true that NASA's mission would probably not support a much greater emphasis in small Earth Venture missions.

[Response] Noted. ESA (and other agencies) are also pursuing smaller missions which we can further highlight. The issue of cost versus quality (i.e. commercial radiometric requirements are probably not as strict those from space agencies) is an important element of this discussion.

---

## Author Comment (AC4) · 19 Apr 2017

First off, the paper is excellent and extremely valuable to the community, and, frankly, thought provoking. I honestly had no idea about more than half of the issues raised in the paper. I can imagine that, for younger scientists/engineers, both hydrologic practitioners and researchers, the paper will be especially motivating. With faculty positions and government research tight, the paper gives some hope for a larger industry including the private sector where many of our current and recent Ph.D. students could find employment, and most importantly, contribute significantly.

[Response]. We thank Prof Salvucci for his kind words and appreciate his review of our manuscript.

The paper is long, but that's ok. It is very well written. I have only a few short comments

that could be considered by the authors before publication:

1) In section 2.1, fundamental challenge #3, where stove-piping is criticized, i wonder if the EOS project isn't a counter-example (at least partly).

[Response]. Noted and it may be worth highlighting such a counter example.

2) The point made in section 2.1, fundamental challenge #4, that due to merging "observed inter-annual fluctuations may reflect discontinuities in the constellation of satellites" is an important one, and thus could use a few citations, as this has been discussed a fair amount, and many merged products exist to attempt to overcome it.

[Response]. This is a good point and some appropriate examples and references can be included.

3) in the future agency missions discussion (3.1), the statement "Deep soil moisture could also be on the list, although soil moisture algorithms that make use of wavelengths longer than L-band are less than mature." is made without evidence. I honestly do not know the status of that research, but I support its potential importance, and am not comfortable with it being dismissed without backing.

[Response]. A suitable reference can be included to support this statement.

4) Two sources of data that I was surprised were not discussed at all (or barely), are: 1) Ameriflux/Fluxnet (I believe there was one citation to a paper that used fluxnet). I believe fluxnet has had a huge impact on hydrologic science, and hope to see it continue and grow in scope. I think it could be argued, as well, that for the cost of a space mission (billion dollars ?), one could put together a pretty amazing network of eddy covariance stations (perhaps 5,000 stations running for 5-10 years ?)

[Response]. Fluxnet is a great example of a community-led approach to sensing that has provided an invaluable source of ground-based data. Within the evaporation community, for instance, it is the gold-standard monitoring network (n.b. we cite three papers in this area). However, the focus of this paper leans more towards emerging

approaches and newer sources of observation data. Fluxnet is a far more mature and well developed technology. The point on the relative cost-allocation of ground-based infrastructure versus space-based missions is an interesting one, and can perhaps be included in a relevant section of the paper. Of course, identifying who funds such an effort is the challenge: it is extremely unlikely that a space agency will seek to allocate their limited resources to such an activity.

2) AMDAR/ACARS observations (e.g. Drue et al, 2008, QUARTERLY JOURNAL OF THE ROYAL METEOROLOGICAL SOCIETY Q. J. R. Meteorol. Soc. 134: 229–239 (2008)) which provide extremely high resolution profiles of temperature, wind and humidity that well sample the atmospheric boundary layer during takeoff and landing many hundreds of times per day all over the planet.

[Response]. Thank you for this citation. We briefly discuss AMDAR in Section 3.6 (Signals of Opportunity) but lacked a published reference.

---

## Author Comment (AC5) · 19 Apr 2017

This paper is an admirably comprehensive overview of how changes in sensing platforms, data providers of spaceborne applications, enable new data sources, and thus, presumably, new breakthroughs in hydrology. No single paper can be expected to comprehensively cover the entire possible 'future of hydrology' and thus this provides a somewhat biased view (about which a caveat in the text would be useful). Nevertheless, the authors have done a remarkably thorough job in terms of capturing the various exciting opportunities available. I particularly appreciate the high number of references across such a diverse paper. Nevertheless, some increased focus and streamlining would significantly improve the paper and is necessary. Many of the points made in the current paper are likely to be lost on the reader without it.

[Response]. We thank the reviewer for their thoughtful comments and obvious attention to detail in reviewing this paper. We agree that there are specific aspects of the paper that may require some refining during the revision process, and your comments will help in directing this activity.

Specific Comments:

1) Section 2 isn't clearly tied to the rest of the paper. Section 2.1 is a haphazard mix of challenges that can be addressed by using a wider diversity of platforms, commercial or otherwise, highlighted in Sections 3 and 4 (e.g. the risk of launches and focusing on single large instruments), and ones that are unrelated (e.g. greater multi-mission integration could be achieved also with space agencies and traditional large LEO or geostationary satellites), and ones that may even be made worse if traditional EO platforms break down (e.g. the perception that remote sensing data are magic and ignorance about reliance on retrieval algorithms may become worse with people using more commercial solutions that, for reasons of competition with other companies, are far more likely to keep their technological and retrieval details secret). Additionally, the challenges and opportunities mentioned for each variable in Section 2.2 are somewhat haphazard. It would be useful to specifically link each of the variables to opportunities mentioned in the rest of the paper.

[Response]. The paper seeks to provide an overview of the current state of EO, offer a perspective on outstanding challenges and knowledge gaps, and highlight the opportunities (and limitations) of emerging sensing platforms that have the potential to advance the EO field. Section 2 is a central element of this purpose, reflecting both on the capabilities for hydrological retrieval, which is deliberately concise and largely satellite focused, as well as identifying some of the broader EO "Problems, Challenges and Knowledge Gaps" (Section 2.1). We would disagree that these are a haphazard collection: the listed challenges, while not exhaustive, are certainly relevant, pertinent and in some instances, present as immediate roadblocks to furthering progress in the field. It is important to clarify that Section 3 and 4 are not presented as solutions to

the challenges and issues outlined in Section 2. We are not advocating that simply providing more observations is the answer to these challenges. As such, providing a direct link to the variable specific issues outlined in Section 2 to these later sections is not the intent or purpose of our work. However, we appreciate the comment that further effort is needed to better integrate the different elements of this paper into a cohesive and self-contained manuscript, and will attempt to address this in a revised version.

2) Section 4 comes across too much as an advertisement for EO start-ups in Silicon Valley and a list of problems with space agency-built sensors (as also mentioned in Josh Fisher's non-reviewer comment). The challenges associated with commercial data are mentioned either as an aside that belies the serious nature of the issue (unknown and most likely lacking calibration relative to scientific standards in most cases), or not at all (data continuity, data costs, the fact that Cubesat orbits aren't always perfectly predictable a priori because of the way in which they are launched). The lower budgets of these startups are presented as being due to the nimbleness of commercial industry, but don't reflect the fact that launching lots of small satellites is inherently cheaper, or that SpaceX has relied on massive amounts of capital investments and has a high failure rate. This is a dis-service not only to government agencies as a source of EO data, but also to the points raised in Section II (which are lost in the process) and to the fact that, in reality, and the fact that future models and analyses efforts are likely to be best aided by a combination of both single-sensor satellites that provide reliable data at perhaps coarser spatial resolution and high resolution data from Cubesats. This section would be significantly aided by a summary table of pros and cons for a) government vs. commercial observing systems. The enormous potential of commercial systems (rightfully highlighted) would go to waste if the community cannot also think and overcome their related challenges. If space is an issue, such a table would therefore be more useful than e.g. table I, which summarizes established knowledge.

[Response]. We thank the reviewer for their perspective on this. As is detailed in our response to Dr Fisher (see http://www.hydrol-earth-syst-sci-discuss.net/hess-

2017-54/hess-2017-54-AC2-print.pdf), we are certainly not seeking to present the commercial sector as the panacea of all problems related to EO. As noted in that response, the opportunities presented by the commercial sector represent just one of a large number of emergent EO technologies that are discussed in our paper, which include advances in UAVs, citizen science, balloons, opportunistic sensing, in addition to important government space-agency driven missions. In terms of the issues associated with commercial systems, we clearly identify calibration limitations (p.36 l. 18-20 and elsewhere) and explicitly refer to the issue of data continuity and data costs (in the title of Section 4.3 as well as throughout p.36 last paragraph and p.37). That commercial are cheaper, is reflected in the type and quality of the product: a point that can perhaps be reiterated. Further effort to highlight these very real challenges can be made in the revised version. However, as we are not advocating an "us versus them" approach to EO between the government and commercial spheres, and considering that the pro's and con's are already well stated throughout Section 4.3, the need for a table will require some consideration. Likewise, we will carefully review the paper for perceived bias, while also being mindful that such pointed discussions of both commercial and agency based EO are somewhat rare in the literature, and while being a potentially sensitive topic, may also represent as an important agent of change.

Relatedly, the authors may also want to consider renaming the paper "Opportunities and challenges for the future of remote sensing" to further highlight that a world where hydrologists predominantly use data from the start-ups and new platforms identified in this paper would be great but is not guaranteed if the challenges mentioned throughout are overcome.

[Response]. We appreciate the reviewers' suggestion and will consider the most appropriate title for the paper upon revision. But again, we are not advocating this particular vision of the future: the best outcome, as the reviewer notes, is likely the one that combines available systems and sensing platforms.

3) Section 3.2 only has one subheading, and it is unclear why UAVs deserve their own

subheading but balloons and solar planes do not.

[Response]. Noted. This was a formatting mistake made during final edits. 3.2 should be UAVs alone and there will be no subsequent sub-heading, as Balloons and Solar Planes are detailed in Section 3.3.

4) It would be useful to have a short summary section somewhere that explicitly summarizes challenges to the community and how it operates. E.g. the need to move towards cloud more, publish processed datasets to the same cloud platform rather than keeping them behind user accounts, think about how to engage with commercial agents, building data ingestion systems that are based on more haphazard temporal and spatial coverage, etc. Such issues are now mentioned as asides throughout and may seem less important, although they are arguably among the most potentially actionable results of the paper.

[Response]. As has been noted by many of the reviewers, the paper is already quite long and exhaustive, so we are reticent to add additional sections to the manuscript. However, as this reviewer rightly states, this may be an important missing element. We have tried to steer away from prescriptive action and instead provide an overview of challenges, limitations and opportunities. Perhaps a paragraph in the conclusion section would suffice to draw attention to these important actions.

5) Several sections have noticeably lower added value than others and may be best deleted to keep the paper length manageable and make the rest of the paper clearer. For example, the historical introduction about ground measurements in Section 3.6 (p. 25) only barely links up to the rest of the section (except maybe through the "Internet of Things" reference, but even that is only useful to someone who has already spent time thinking about the IoT and will grasp the connection immediately). Similarly, discussion of the declining trend in ground observations in Section 2.1 breaks the flow and is somewhat out of place. The section on potential new airborne measurements for commercial aircraft (p. 27) is far too vague – what variables could possibly be easily

added? Airborne instrumentation for radar, fluorometry or even optical measurements requires extensive retrofitting of the aircraft and e.g. instrument protection in the form of a radome. By contrast eddy covariance measurements are difficult to calibrate.

[Response]. Noted. We will attempt to streamline some of these discussions while maintaining the intent of the underlying message. The decline of in-situ networks (Sect 2.1) is an important issue that often gets lost in the excitement of new satellite missions and emerging technologies: we will consider how best to phrase and place this in the context of the paper; likewise the introduction to Section 3.6. In terms of airborne sensing, this is a deliberately brief section that has some pertinent examples (the AMDAR system), which we will provide a clearer reference to. There are also efforts to install IR based sensors as early warning systems that have a range of potential applications (see Nature 502, 422–423, 2013 doi:10.1038/502422a): but the reviewer is correct in that there are challenges to do this, which can also be mentioned.

Minor Specific Comments:

* Section 3 title: It would be useful to include the phrase "data sources" or "platforms" somewhere in this section title, as that is the focus of many of the sub-headings.

[Response]. Noted and we will examine this in the revision of the paper.

* Page 4, Line 9: The A-Train efforts and large number of sensors aboard Aqua have been useful so some care with phrasing is needed here.

[Response]. Noted, and an excellent example that should be highlighted.

* Page 7, Point 4: The first sentence here provides far too limiting a view of the possible uses of hydrological data. It does not include, for example, the use of hydrologic earth observation data (without further assimilation) in carbon-cycle or biogeochemical studies, epidemiological ones, or even ones on social science implications of variations in water availability (e.g. Muller et al, PNAS, 2016 "Impact of the Syrian refugee crisis on land use and transboundary freshwater resources"). It would be more correct to delete

the first sentence and rephrase that the ability to have long-term trends is useful for many studies instead of being so definitive.

[Response]. We agree and the reviewer has provided several appropriate examples to broaden this perspective.

* Page 7, Line 19: Su et al, Geophysical Research Letters, 2016 ("Homogeneity of a global multi-satellite soil moisture climate data record") may be a helpful citation in the discussion of multi-sensor merging

[Response]. Noted and we appreciate the reference to highlight this important aspect of research.

* Page 12, Line 3: Note that e.g. Konings and Gentine, Global Change Biology, 2017 ("Global variations in ecosystem-scale isohydricity") is a notable exception here.

[Response]. Thanks for providing this interesting reference.

* Page 15, line 14: NASA/DLR rather than NASA/German

[Response]. Noted

* Page 36, Line 13: The Chinese WCOM mission may be worth mentioning as an exception here, though it is still in the early phases

[Response]. Noted. WCOM is discussed on pg. 15 lines 20-26. While this is a strong hydrological focused mission proposal, it does not cover the full range of variables needed for closure.

* You may want to mention that Planet was recently sold

[Response]. To our knowledge, Planet has not been sold. They acquired Blackridge (operators of RapidEye) and are negotiating the purchase of Google's Terra Bella satellite imaging capability.

* The 220 MB multimedia supplement may not be worth its large file size. Although it is

difficult to note the cars moving unless looking for it. If the authors really want to include it, it may be worth cropping to a smaller area that focuses more on the highways.

[Response]. Noted. We will consider the value of this supplementary material.

* A few informal abbreviations have slipped in throughout: tech rather than technology, 'till, jet Propulson lab rather than laboratory, etc..

[Response]. Noted. A careful editing will ensure these are removed during the revision process.

---

## Author Response (AR1)

Dear Prof Lettenmaier.

Below you will find our individual response to each of the reviewer and short comment contributions. We have implemented the majority of the suggested changes, and where we have not, a justification has been provided. A particular effort was made to better link some of the Sections to each other, so that the manuscript provided a more logical development of the ideas and emerging EO technologies that are discussed therein, and to streamline the text.

Overall, we believe that the revised manuscript represents a much stronger contribution as a result of this process and we would take this opportunity to again thank the reviewers for their time in providing a constructive assessment of our manuscript.

Best wishes,

Matthew McCabe

**Response to Anonymous Reviewer.**

**Specific Comments:**

**1) Section 2 isn't clearly tied to the rest of the paper. Section 2.1 is a haphazard mix of challenges that can be addressed by using a wider diversity of platforms, commercial or otherwise, highlighted in Sections 3 and 4 (e.g. the risk of launches and focusing on single large instruments), and ones that are unrelated (e.g. greater multi-mission integration could be achieved also with space agencies and traditional large LEO or geostationary satellites), and ones that may even be made worse if traditional EO platforms break down (e.g. the perception that remote sensing data are magic and ignorance about reliance on retrieval algorithms may become worse with people using more commercial solutions that, for reasons of competition with other companies, are far more likely to keep their technological and retrieval details secret). Additionally, the challenges and opportunities mentioned for each variable in Section 2.2 are somewhat haphazard. It would be useful to specifically link each of the variables to opportunities mentioned in the rest of the paper.**

[Response]. The paper seeks to provide an overview of the current state of EO, offer a perspective on outstanding challenges and knowledge gaps, and highlight the opportunities (and limitations) of emerging sensing platforms that have the potential to advance the EO field. Section 2 is a central element of this purpose, reflecting both on the capabilities for hydrological retrieval, which is deliberately concise and largely satellite focused, as well as identifying some of the broader EO "Problems, Challenges and Knowledge Gaps" (Section 2.1). We would disagree that these are a haphazard collection: the listed challenges, while not exhaustive, are certainly relevant, pertinent and in some instances, present as immediate roadblocks to furthering progress in the field. It is important to clarify that Section 3 and 4 are not presented as solutions to the challenges and issues outlined in Section 2. We are not advocating that simply providing more observations is the answer to these challenges. As such, providing a direct link to the variable specific issues outlined in Section 2 to these later sections, is not the intent or purpose of our work. However, we appreciate the comment that further effort is needed to better integrate the different elements of this paper into a cohesive and self-contained manuscript, and have attempted to address this throughout. We have restructured Section 2.2 with specific headings and have added an introductory statement in Section 4 to link back to these earlier sections. Where relevant, we have made explicit links to different sections through the Section 2 text.

**2) Section 4 comes across too much as an advertisement for EO start-ups in Silicon Valley and a list of problems with space agency-built sensors (as also mentioned in Josh Fisher's non-reviewer comment). The challenges associated with commercial data are mentioned either as an aside that belies the serious nature of the issue (unknown and most likely lacking calibration relative to scientific standards in most cases) , or not at all (data continuity, data costs, the fact that Cubesat orbits aren't always perfectly predictable a priori**

**because of the way in which they are launched). The lower budgets of these startups are presented as being due to the nimbleness of commercial industry, but don't reflect the fact that launching lots of small satellites is inherently cheaper, or that SpaceX has relied on massive amounts of capital investments and has a high failure rate. This is a dis-service not only to government agencies as a source of EO data, but also to the points raised in Section II (which are lost in the process) and to the fact that, in reality, and the fact that future models and analyses efforts are likely to be best aided by a combination of both single-sensor satellites that provide reliable data at perhaps coarser spatial resolution and high resolution data from Cubesats. This section would be significantly aided by a summary table of pros and cons for a) government vs. commercial observing systems. The enormous potential of commercial systems (rightfully highlighted) would go to waste if the community cannot also think and overcome their related challenges. If space is an issue, such a table would therefore be more useful than e.g. table I, which summarizes established knowledge.**

[Response]. We thank the reviewer for their perspective on this. As is detailed in our response to Dr Fisher, we are certainly not seeking to present the commercial sector as the panacea of all the problems related to EO. As noted in that response, the opportunities presented by the commercial sector represent just one of a large number of emergent EO technologies that are discussed in our paper, which include advances in UAVs, citizen science, balloons, opportunistic sensing, in addition to important government space-agency driven missions. In terms of the issues associated with commercial systems, we clearly identify calibration limitations (p.36 l. 18-20 and elsewhere) and explicitly refer to the issue of data continuity and data costs (in the title of Section 4.3 as well as throughout p.36 last paragraph and p.37). That commercial are cheaper, is reflected in the type and quality of the product: a point that can perhaps be reiterated. As we are not advocating an "us versus them" approach to EO between the government and commercial spheres, and considering that the pro's and con's are already well stated throughout Section 4.3, we have not included a separate table. We have carefully reviewed the paper for perceived bias. We are also mindful that such pointed discussions of both commercial and agency based EO are somewhat rare in the literature and offer an opportunity to present a variety of perspectives on this important issue.

**Relatedly, the authors may also want to consider renaming the paper "Opportunities and challenges for the future of remote sensing" to further highlight that a world where hydrologists predominantly use data from the start-ups and new platforms identified in this paper would be great but is not guaranteed if the challenges mentioned throughout are overcome.**

[Response]. We have decided to maintain the original title.

**3) Section 3.2 only has one subheading, and it is unclear why UAVs deserve their own subheading but balloons and solar planes do not.**

[Response]. Thanks for noting this mistake: we have removed Section 3.2.1 and changed the sub-headings for both Section 3.2 and 3.3.

**4) It would be useful to have a short summary section somewhere that explicitly summarizes challenges to the community and how it operates. E.g. the need to move towards cloud more, publish processed datasets to the same cloud platform rather than keeping them behind user accounts, think about how to engage with commercial agents, building data ingestion systems that are based on more haphazard temporal and spatial coverage, etc. Such issues are now mentioned as asides throughout and may seem less important, although they are arguably among the most potentially actionable results of the paper.**

[Response]. Unfortunately, as has been noted by many of the reviewers, the paper is already quite long and exhaustive, so we are reticent to add further sections to the manuscript. However, these important points are well stated throughout the relevant sections.

**5) Several sections have noticeably lower added value than others, and may be best deleted to keep the paper length manageable and make the rest of the paper clearer. For example, the historical introduction about ground measurements in Section 3.6 (p. 25) only barely links up to the rest of the section (except maybe through the "Internet of Things" reference, but even that is only useful to someone who has already**

**spent time thinking about the IoT and will grasp the connection immediately). Similarly, discussion of the declining trend in ground observations in Section 2.1 breaks the flow and is somewhat out of place. The section on potential new airborne measurements for commercial aircraft (p. 27) is far too vague – what variables could possibly be easily added? Airborne instrumentation for radar, fluorometry or even optical measurements requires extensive retrofitting of the aircraft and e.g. instrument protection in the form of a radome. By contrast eddy covariance measurements are difficult to calibrate.**

[Response]. We have now removed the introduction in Section 3.6, starting instead with an immediate introduction to the IoT. In terms of the airborne instrumentation, apart from expanding on the AMDAR system, which is already used operationally e.g. (Petersen, 2016), we have included another reference to research looking at integrated systems that measure atmospheric turbulence (Sharman et al., 2014), as well as for volcanic ash detection (Prata et al., 2016). We have also highlighted some of the challenges in implementing such an observing system.

**Minor Specific Comments:**

**\* Section 3 title: It would be useful to include the phrase "data sources" or "platforms" somewhere in this section title, as that is the focus of many of the sub-headings.**

[Response]. The section title has been changed to "Emergent Platforms, Capabilities and Technologies"

**\* Page 4, Line 9: The A-Train efforts and large number of sensors aboard Aqua have been useful so some care with phrasing is needed here.**

[Response]. Noted, and an excellent example that should be highlighted. We have adjusted the text such that it now reads "However, attempts to coordinate multi-platform observing systems in recognition of shared goals such as holistic water cycle measurement have been less effective: although NASA's A-train may serve as a partial counter-example to this claim (Stephens et al., 2002)."

**\* Page 7, Point 4: The first sentence here provides far too limiting a view of the possible uses of hydrological data. It does not include, for example, the use of hydrologic earth observation data (without further assimilation) in carbon-cycle or biogeochemical studies, epidemiological ones, or even ones on social science implications of variations in water availability (e.g. Muller et al, PNAS, 2016 "Impact of the Syrian refugee crisis on land use and transboundary freshwater resources"). It would be more correct to delete the first sentence and rephrase that the ability to have long-term trends is useful for many studies instead of being so definitive.**

[Response]. We agree and the reviewer has provided several appropriate examples to broaden this perspective. The introductory sentence has been adjusted and now reads as "The capacity to develop long-term remotely sensed hydrologic records have proven useful across a range of applications, including: (a) studying trends within the terrestrial water cycle (Miralles et al., 2014), (b) improving simulations in hydrological, eco-physiological or biogeochemical models, (c) examining the social science implications of water availability (Müller et al., 2016), or (d) benchmarking the hydrology in land-surface and atmospheric models that project the impact of climate and land use change on the water cycle (Mueller et al., 2013), amongst numerous other examples."

**\* Page 7, Line 19: Su et al, Geophysical Research Letters, 2016 ("Homogeneity of a global multi-satellite soil moisture climate data record") may be a helpful citation in the discussion of multi-sensor merging**

[Response]. Noted and we appreciate the reference to highlight this important aspect of research. (Su et al., 2016) has now been included at the end of this paragraph.

**\* Page 12, Line 3: Note that e.g. Konings and Gentine, Global Change Biology, 2017 ("Global variations in ecosystem-scale isohydricity") is a noteable exception here.**

[Response]. Thanks for providing this interesting reference. While a very interesting paper, it does not relate to the use of SIF measurements in particular (the focus of this paragraph), but uses VOD values instead.

**\* Page 15, line 14: NASA/DLR rather than NASA/German**

[Response]. Noted and now changed to "The joint NASA and German Aerospace Center (DLR) GRACE Follow-On"

**\* Page 36, Line 13: The Chinese WCOM mission may be worth mentioning as an exception here, though it is still in the early phases**

[Response]. Noted. WCOM is discussed on pg. 15 lines 20-26. While this is a strong hydrological focused mission proposal, it does not cover the full range of variables needed for closure.

**\* You may want to mention that Planet was recently sold**

[Response]. To our knowledge, Planet has not been sold. They acquired Blackridge (operators of RapidEye) and have recently finalized the purchase of Google's TerraBella space imaging company.

**\* The 220 MB multimedia supplement may not be worth its large file size. Although it is difficult to note the cars moving unless looking for it. If the authors really want to include it, it may be worth cropping to a smaller area that focuses more on the highways.**

[Response]. We have retained this as supplementary material.

**\* A few informal abbreviations have slipped in throughout: tech rather than technology, 'till, jet Propulson lab rather than laboratory, etc..**

[Response]. We removed "tech" from Silicon Valley tech companies (pg. 19 line 24) and changed Lab to Laboratory (pg 21. line 12).
* * *
**Response to Prof Kummerow**

**p. 7, line 10: The notion that we need a more appropriate error analysis before we can merge multiple sensors into a coherent framework is often repeated and yet to be implemented. Since this is a large part of what the paper is advocating for the future, the authors should emphasize the critical need to advance this field rather than simply stating it as a necessity.**

The specific purpose of Section 2.1 is to highlight some of the challenges and knowledge gaps in remote sensing that require community attention. The particular challenge of multi-sensor integration and error characterization is a key example of these challenges. We have adjusted this paragraph to explicitly state the need, by adjusted the text to read "Efforts to harmonize satellite data **represent a critical need**, not just for more effective data assimilation or direct use of Earth observations in hydrological models, but to better understand any implied signal of trend or variability in derived variables."

**p. 29, line 22: Google's Earth Engine was developed by Google and follows Google's rules. Without a coordinating body in global hydrology, it becomes difficult to reproduce something like this.**

Agreed. It is also difficult to imagine the scientific community organizing themselves and then funding an initiative to replicate such a framework. Whether it is necessary to do so, given that these systems are relatively easy to access and (currently) not especially restrictive, is also a consideration. There are examples of private-public partnerships that seek to reproduce some of the same functions as Google Earth Engine and Amazon Web Server (see discussion on EODC p.30, line 15) which may serve as a model should these commercial facilities prove unsuitable for scientific application. We have updated the text here to include the following statement "Where this cloud-computing might take place also raises questions (and potential concerns), related to data-archiving, distribution and intellectual property."

**p. 34, line 27: The fallacy here is that it assumes that if SpaceX develops a very cheap launch vehicle, then NASA would launch its satellites on it. That is probably not the case. NASA would continue to develop new technologies that in its early stages, might be just as expensive as before.**

Hopefully NASA will continue to develop new technologies, as the commercial sector are unlikely to invest the significant resources required to do this: a point mentioned in our paper. However, in terms of delivering satellites into space and re-provisioning ISS, NASA is already utilizing commercial sector expertise, and this model seems likely to expand. But the risk of duplication of effort is also apparent.

**p. 30, line 22: Storage can get very large. While commercial vendors often discard old data which generates little revenue, this has never been the case in Earth Sciences. It again goes back to the question of funding data storage that belongs to the community rather than an "agency".**

Yes, this is an important point, and there are examples within our own scientific communities where this has been done poorly. We have updated the text to reflect the importance of this issue "Earth observation data archiving and stewardship are relatively new concepts to these more commercial oriented services, so it is unclear how effectively they will embrace the scientific model of data retention: especially if the revenue potential of older data does not justify its storage. Whether agencies will continue to maintain their own storage services or leverage these much larger commercial facilities also remains to be seen. Regardless of any future delivery mode, ensuring continued free-access and long-term archiving of stored earth observations is essential to advancing the field. With the rise of artificial intelligence and deep learning approaches (discussed below), the importance of maintaining a long record of "training data" may provide the incentive to archive historical earth observations."

**Abstract, line 10: The real time, high resolution video capabilities are surely new, but equally important for cloud development are the newly available JMA and NOAA 1min. geostationary data with 1 km resolution. CMA even has 60m resolution geostationary data with 30 sec. resolution. Given the size and speed with which clouds naturally evolve, it is not clear to me that 10 m data with 30 frames per second would revolutionize our understanding of clouds. I can better see the utility in tracking small scale, fast flows. In any case some careful rewording here would help.**

We have revised this text to remove the cloud formation statement. It now reads as "With these advances come new space-borne measurements, such as real-time high-definition video for monitoring air pollution, storm-cell development, flood propagation, precipitation tracking, or even for constructing digital surfaces using structure-from-motion techniques."

**p. 6, line 5: Eliminate "radiative" – just visible and NIR frequencies is enough.**

Done

**p. 6, line 6: It is now GOES-16 and authors should probably mention Himawari-8 as well. They do the same thing.**

Done

**p. 9, Line 16: Doppler radars are good for judging speeds and rotation of convective elements. They don't do anything for rainfall estimation. I suggest that the author drop "Doppler" or replace it with "polarimetric".**

Done. We have removed the reference to Doppler here and replaced with polarimetric, or removed it entirely to just mention "radars".

**p. 14, Line 5: Section 2 discussed variables independently. The author mentions that integration and a more holistic approach is necessary – but emergent capabilities and technologies are not the answer. They merely add observations. The text is not very clear here.**

We agree that more observations will not directly resolve the issues raised in Section 2. The purpose of Section 3 (Emergent Capabilities and Technologies) is not to provide answers to these issues. We have rephrased this

paragraph to highlight the importance of exploiting a comprehensive strategy that includes both current and future data sources: "In order to drive continued advances in our system understanding, it is paramount that we exploit a comprehensive and holistic EO strategy, both with the data that we currently have, as well as that which is only just emerging."

**p. 14, line 11: The text starts out boldly predicting hydrology 50 years from now. The paper, however, focuses on things that are real today and probably represents the next 10-15 years. Perhaps that would be a better into?**

The introductory line is definitely not intended to impart any bold or long term predictions of hydrology. Perhaps the comparison to the developments and improvements in our transportation system is not the clearest analogy. We will review this. Our focus is certainly on the near-term (5-10 year horizon), highlighting technologies that for the most part have emerged within the last few years, but which we believe have the potential to play a much larger role in EO strategies and in advancing the EO landscape. We have changed the text to reflect this "A few decades from now, historians may reflect on today's remote sensing capabilities the way we regard transportation in the early 20th century, i.e. most of the major modes were already in existence, but huge improvements in quality, cost and production efficiency, accessibility, and safety were yet to come."

**p. 14: Line 33: Perhaps Japan should be mentioned as well.**

Noted and included

**p. 18, line 25: The balloon section needs a few more caveats. Since high altitude balloons have been around for a long time, and the authors fail to make a case that commercial balloons for telecom would somehow change their utility or cost, some argument for why we might see a resurgence of balloon-based observing systems should be put forth.**

The argument here is that commercial enterprises (i.e. Google and Facebook) have active programs exploring the use of maneuverable high-altitude balloons and/or autonomous planes. This is driven by commercial motives for the most part. Whether our community can leverage this for scientific purposes remains unclear. We have further emphasized balloons by adding "Harnessing a fleet of high-altitude balloons or aircraft…" as well as indicating the commercial drivers "Leveraging the advances in technology behind the commercial development and ultimate production…" on page 20.

**p. 18, line 32: "constructed" instead of "constructing".**

Noted

**p. 21, line 19: perhaps the authors need to still remind readers that while passive systems in the VIS/IR are quite feasible, the power needed by active sensors still limits these systems today.**

Noted. We have updated the text to read "Regardless of the driving forces behind their emergence, CubeSats represent a cost-effective observation strategy that provides a unique opportunity for the implementation and demonstration of technological innovations, serving as potential test beds for advanced visible-infrared sensing systems (the power requirements of active sensors currently limit their integration to date)…"

**p. 23, line 5: I think the authors could come up with a better example here. Rain gauge records are often protected for various geopolitical reasons. It is not clear to me that the rain gauge records from the Congo have not been released simply because nobody in the office has a smartphone.**

We have rewritten this paragraph to include a number of new examples and references "…while the delivery of information via text-messaging has improved the economic outcomes of subsistence farmers through simple knowledge of market prices (Wyche and Steinfield, 2016). Other approaches exploit their image capturing capabilities combined with smartphone applications to monitor soil, vegetation and land use changes (Herrick et al., 2017) or flash floods and river stage (Le Coz et al., 2016). In this sense, a person with a smartphone becomes a remote, or at least proximal, sensing platform capable of providing information on the environment

around them. This concept of harnessing widely accessible technology and the users deploying it is broadly referred to as "citizen science", and has the potential to reshape how information is both collected and interpreted (Buytaert et al., 2014)."

Additional examples are included in later paragraphs.

**p. 24, line 12: The authors might want to include CoCoRAHs https://www.cocorahs.org/ Which, on a given day, receives roughly 7000 daily rain gauge reports) from citizen scientists.**

An excellent example of 20,000 participants. We have updated the text to include this "A larger scale example includes the Community Collaborative Rain, Hail and Snow Network in the United States, (https://www.cocorahs.org/), which receive approximately 20,000 daily rain-gauge reports from citizen scientists across North America (Reges et al., 2016)."

**p. 25, line 20: Perhaps it is worth noting that the "entire country" was the Netherlands and that this would not work for countries like the US where many rural areas do not have nearly enough cell phone towers to apply this technique.**

We note that there are a number of other examples for developing country applications of this approach, as is described here http://www.nature.com/news/mobile-phone-signals-bolster-street-level-rain-forecasts-1.21799

**p. 27, line 3: Commercial aircraft started making turbulence measurements as well. See Sharman et al., Description and Derived Climatologies of Automated In Situ Eddy Dissipation-Rate, 2014. Appl. Met. and Clim., 53, 1416-1432**

Noted and this example has been included in the text.

**p. 27, line 25. The same issue pointed out in the abstract.**

We would note that the geostationary systems mentioned do not provide HD video: the focus of this section. We have however adjusted the text to remove the focus on reshaping our knowledge of cloud processes.

**p. 33, line 4: The risk averse nature of space agencies is true. The rationale is the authors' interpretation. I would argue that they are risk averse because it takes so long to reach scientific consensus. The PI led missions have very short time frames – similar to industry. Perhaps pointing out that facility type missions are risk averse while PI missions can be much more nimble is a better way of expressing this.**

Noted.

**p. 35, line 20: At least NASA is shifting into smaller missions via Earth Ventures. This should be acknowledged although it is equally true that NASA's mission would probably not support a much greater emphasis in small Earth Venture missions.**

We mention the NASA Venture missions at the beginning of Section 3.1 "While NASA Venture class and smaller missions as well as bolt-on instruments typically have more compressed timelines, it is clear that the time horizons from mission concept to launch are long, rather than short" and elsewhere in that section.
* * *
**Response to Interactive Comment by Josh Fisher.**

The key criticism raised by Dr Fisher is that our paper is somehow framed in an "*us-versus-them*" context. Given the strong links that have long existed between the commercial and government sectors (and further evidenced by recent joint-ventures such as rocket launches, space-station supply and equipment delivery, sensor design-and-build partnerships etc.) an "*us-versus-them*" position would not be a strong one to advocate. Likewise, we are certainly not seeking to prosecute any particular case through this manuscript, nor is there any desire to act as 'marketing agents' to the commercial sector. Regardless, we will certainly review the manuscript to assess any claim of perceived bias closely. We would note that commercial satellite systems represent just one

(relatively small) facet of a number of exciting EO opportunities that are discussed throughout the paper. Others include: a) advances in UAVs; b) citizen science; c) stratospheric balloons; d) opportunistic sensing; e) CubeSats; and of course f) government-driven satellite systems, which form the backbone of our EO capability. Given that the entirety of Section 3 (which comprises approximately 50% of the paper) is focused on detailing the emergence of these new approaches to EO, we can only assume that the suggested title change has been made somewhat facetiously.

In our assessment of the "Changing Earth Observation Landscape" detailed throughout Section 4, we seek to provide a rational assessment of traditional space agency approaches in the light of some recent (and planned) commercial sector activities. We do not focus this comparison through any particular lens, or seek to convey some preconceived bias, but rather offer a statement of documented experiences together with supporting evidence. That the commercial and government sectors may be approaching the delivery of satellites and sensors in different ways is a function of many contributing elements, most of which are detailed in the paper. Nowhere do we suggest that there is an equivalence of launched sensors: indeed, we clearly identify and highlight some of the limitations of commercial activities, along with areas of potential concern. While the opportunities being afforded by the emergence of commercial platforms are exciting and potentially game-changing, there are very clear scientific and access issues that will need to be addressed for these to impact on research-based EO investigations. As is noted by Dr Fisher, these are well discussed and described: but it seems that further effort needs to be made to draw attention to these issues in the text.

We agree with Dr Fisher that detailing "*the key science and applications needs, the necessary observational requirements, the associated technological capabilities, and then map those onto commercial capabilities*" is something that is "*desperately needed*". However, this is neither the intent or purpose of our manuscript, which is described quite concisely in the abstract and encapsulated well within the title. Fortunately, the much anticipated National Academies Earth Science and Applications from Space Decadal Survey, to be released later in 2017, will likely cover many of these suggested topics.

We look forward to incorporating some of Dr Fisher's thoughtful comments in a revised manuscript.
* * *
**Response to Prof Salvucci (Referee)**

**1) In section 2.1, fundamental challenge #3, where stove-piping is criticized, i wonder if the EOS project isn't a counter-example (at least partly).**

[Response]. Unfortunately, there are a number of stove-piping examples within the EOS program that limit this as an unfettered counter-example.

**2) The point made in section 2.1, fundamental challenge #4, that due to merging "observed inter-annual fluctuations may reflect discontinuities in the constellation of satellites" is an important one, and thus could use a few citations, as this has been discussed a fair amount, and many merged products exist to attempt to overcome it.**

[Response]. We have added a number of references to this point following our response to the anonymous reviewer (see above).

**3) in the future agency missions discussion (3.1), the statement "Deep soil moisture could also be on the list, although soil moisture algorithms that make use of wavelengths longer than L-band are less than mature." is made without evidence. I honestly do not know the status of that research, but I support its potential importance, and am not comfortable with it being dismissed without backing.**

[Response]. Unfortunately, mature examples of using longer than L-Band wavelengths for soil moisture retrieval from space could not be found in the literature. I have included a reference to a mission proposal for an L- and P-Band active system (Moghaddam et al., 2007) in the text.

**4) Two sources of data that I was surprised were not discussed at all (or barely), are :**

**1) Ameriflux/Fluxnet (I believe there was one citation to a paper that used fluxnet). I believe fluxnet has had a huge impact on hydrologic science, and hope to see it continue and grow in scope. I think it could be argued, as well, that for the cost of a space mission (billion dollars ?), one could put together a pretty amazing network of eddy covariance stations (perhaps 5,000 stations running for 5-10 years ?)**

[Response]. Fluxnet is a great example of a community led approach to sensing that has provided an invaluable source of ground-based data. Within the evaporation community for instance, it is the gold-standard monitoring network (n.b. we cite three papers in this area). However, the focus of this paper leans more towards emerging approaches and newer sources of observation data. Fluxnet is a far more mature and well developed technology. The point on the relative cost-allocation of ground based infrastructure versus space-based missions is an interesting one, and can perhaps be included in a relevant section of the paper. Of course, identifying who funds such an effort is the challenge: it is extremely unlikely that a space agency will seek to allocate their limited resources to such an activity.

**2) AMDAR/ACARS observations (e.g. Drue et al, 2008, QUARTERLY JOURNAL OF THE ROYAL METEOROLOGICAL SOCIETY Q. J. R. Meteorol. Soc. 134: 229–239 (2008)) which provide extremely high resolution profiles of temperature, wind and humidity that well sample the atmospheric boundary layer during takeoff and landing many hundreds of times per day all over the planet.**

[Response]. Thank you for this citation (Drüe et al., 2008). We briefly discuss AMDAR in Section 3.6 (Signals of Opportunity) but lacked a published reference. We have now included two additional and recent references to this discussion.

**References**

[revised manuscript text omitted]